behaviour/evolution/genetics

pace-of-life syndrome, personality, heritability, genetic correlations, animal model, primate

**Author for correspondence:**
Pauline B. Zablocki-Thomas
e-mail: pauline.thomas90@gmail.com

# Heritability and genetic correlations of personality, life history and morphology in the grey mouse lemur (*Microcebus murinus*)

Pauline B. Zablocki-Thomas[1,2], Anthony Herrel[1,3], Caitlin J. Karanewsky[4], Fabienne Aujard[1] and Emmanuelle Pouydebat[1]

[1]UMR CNRS/MNHN 7179, Département Adaptations du Vivant, Muséum National d'Histoire Naturelle, Paris, France
[2]Départment de Biologie, École normale supérieure de Lyon, Lyon France
[3]Evolutionary Morphology of Vertebrates, Ghent University, Gent, Belgium
[4]Department of Biochemistry, Stanford University, Stanford, CA, USA

PBZ-T , 0000-0002-2372-4760; AH, 0000-0003-0991-4434; EP, 0000-0002-0542-975X

The recent interest in animal personality has sparked a number of studies on the heritability of personality traits. Yet, how the sources variance these traits can be decomposed remains unclear. Moreover, whether genetic correlations with life-history traits, personality traits and other phenotypic traits exist as predicted by the pace-of-life syndrome hypothesis remains poorly understood. Our aim was to compare the heritability of personality, life-history and morphological traits and their potential genetic correlations in a small primate (*Microcebus murinus*). We performed an animal model analysis on six traits measured in a large sample of captive mouse lemurs (*N* = 486). We chose two personality traits, two life-history traits and two morphological traits to (i) estimate the genetic and/or environmental contribution to their variance, and (ii) test for genetic correlations between these traits. We found modest narrow-sense heritability for personality traits, morphological traits and life-history traits. Other factors including maternal effects also influence the sources of variation in life-history and morphological traits. We found genetic correlations between emergence latency on the one hand and radius length and growth rate on the other hand. Emergence latency was also genetically correlated with birth weight and was influenced by maternal identity. These results provide insights into the influence of genes and

maternal effects on the partitioning of sources of variation in personality, life-history and morphological traits in a captive primate model and suggest that the pace-of-life syndrome may be partly explained by genetic trait covariances.

## 1. Introduction

In his most famous publication, Charles Darwin noted how selection on dogs and pigeons led to the appearance of correlations between traits [1]: hairless dogs frequently showing tooth problems and pigeons with feathered legs having skin between their toes. Furthermore, the famous fox farm experiments showed that selection on behavioural traits may induce correlated evolution on morphological traits [2], suggesting that genetic trait correlations underlie these trait changes. A general theoretical framework to understand the relationship between various phenotypic traits and life-history traits is the pace-of-life syndrome [3], in which behaviour and personality have been recently included [4,5]. This concept links phenotypic traits based on a slow–fast continuum in the 'pace-of-life' of the organism. This is defined by life-history traits, as such as life span, age at sexual maturity or the number of offspring, which are all correlated with one another [3,6–8]. Moreover, individuals are expected to present a bolder and more active personality and a higher metabolic rate and growth rate in relation to a fast pace-of-life (e.g. small rodents). Under this theory, behaviour and life history could be mediated by hormonal determinants [3], yet may also be linked in other ways.

Correlated evolution between different phenotypic traits may arise through genetic correlation [9]. Indeed, traits can evolve together, as, for example, in lizards (*Zootoca vivipara*), where exploration behaviour and resting metabolic rate are correlated, as they present an advantage when they vary in the same direction [10]. Moreover, a recent study showed that by artificially selecting bold and shy lines of zebra fish (*Danio rerio*), the morphology and locomotor performance of these individuals also changed [11]. Another study showed that genetic correlations existed between two personality traits, sociability and boldness, and morphological traits including body pigmentation and size, leading to the apparition of adapted phenotypes that combine several traits [12]. The study of trait correlations is thus an important first step in the understanding of the evolution of traits and trait variation, and may, in part, underlie the trait correlations observed in the pace-of-life syndrome. Falconer & Mackay [13] advocated that the relationship between traits may allow us to detect: (i) the effect of pleiotropic genes, (ii) correlated responses to selection between traits, and (iii) the relationship between the trait and associated fitness.

Personality is generally defined as a statistically repeatable behaviour across context and over time between individuals [14]. These consistent behavioural differences between individuals have been shown to be an important component of individual fitness. A number of studies have focused on personality over the last decades, showing its correlates with other traits like morphology [15] or cognition [16,17], and its heritability in several taxa [18–22]. Some personality traits (e.g. high aggressiveness) have been observed in associations with others (e.g. high boldness) in various taxa [23]. These relationships between behaviours could be induced by common determinants of behaviour, such as hormonal levels, yet can also have underlying genetic determinants.

The narrow-sense heritability of a phenotypic trait is dependent on the presence of additive genetic variation according to its definition [13], which is the ratio of additive genetic variance over the total phenotypic variance. However, the sources of variation in the different types of phenotypic traits are not equal: indeed, the research has shown that fitness traits generally have low heritability, whereas morphological traits have high heritability, and that the heritability of behaviour lies somewhere in between [24]. For life-history traits, selection is probably high and the variability in the additive gene effect ($V_a$) is low as compared with the variability in the phenotype ($V_p$). For morphological traits, selection is typically lower and the variability of additive genetic effects is higher than phenotypic variability [25,26]. Studying and quantifying the heritability of traits and their genetic correlations helps to understand the sources of individual variation and the relative importance of genes in driving phenotypic variation [27,28].

The aim of our study was to describe and compare the heritability of behavioural, life-history and morphological traits and their potential genetic correlations in a small captive primate, *Microcebus murinus* from a captive population [29]. *Microcebus murinus* is a short-lived primate and thus ideal to explore variation in traits as predicted by the pace-of-life syndrome. The captive population used in this study is unique by its size (about 500 individuals) and by the information on life-history

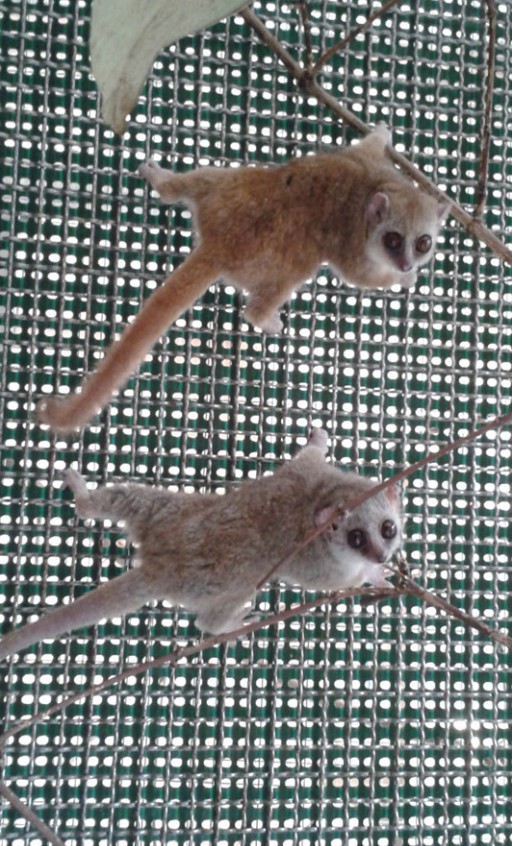

**Figure 1.** Photograph of captive grey mouse lemurs in the Brunoy colony.

and other traits available for these individuals [30] allowing us to explore trait covariation. Information about matrilineal lines are also available since individuals are followed for their entire life. In particular, we aim to explore links between life-history and personality traits under the scope of the 'pace-of-life' hypothesis. Personality traits have been previously quantified for *M. murinus* and shown to be repeatable in both wild and captive animals [15,31], making this an excellent study system. Moreover, phenotypic correlations have been recently described between personality and morphological and life-history traits [15,32], raising the question on the potential genetic bases of these phenotypic covariations.

The aim of this study is to (i) estimate key variance components of morphological and personality traits and compare their heritability to better understand their evolutionary path [33]. We predicted higher heritability for morphological traits compared with behavioural traits, and a low heritability for life-history traits [34–36]. We also aim to (ii) explore genetic correlations between traits to better understand the relationships between life history and behaviour. We expected to find stronger genetic correlations among life-history traits than among morphological traits, with intermediate correlations being expected for behavioural traits [36,37], as well as correlations between life history and personality, as predicted by the pace-of-life syndrome hypothesis.

# 2. Material and methods

## 2.1. Subjects and colony management

We collected data for 486 different grey mouse lemurs (*M. murinus*, figure 1) aged from 1 to 10 years and present in the captive colony of Brunoy (Muséum National d'Histoire Naturelle), originated from eight maternal lines. Individuals are housed in large cages in monosexual groups of three or four individuals. Ambient air temperature is maintained at 25°C and humidity is stable around 30%. All individuals are fed ad libitum, weighed monthly and maintained under artificial light conditions mimicking natural seasons.

## 2.2. Pedigree construction

In captivity, grey mouse lemurs can live up to 12 years, reach sexual maturity in their first year, and females can raise one to four offspring each year [30]. Thus, up to seven generations were present in our pedigree. During the reproductive season, groups of three males and three females from distinct matrilineal origins are placed together for three weeks to mimic the polyandrous mating system [29]; after mating, mothers are isolated and raised with their offspring. As such, all mothers are known and the number of fathers is limited.

We collected DNA samples for 256 individuals and extracted DNA from ear or skin tissue samples (Invitrogen PureLink Genomic DNA mini Kit) and amplified it (Qiagen REPLI-g Mini Kit). The genetic analysis was conducted to determine the paternity for 111 infants for which we disposed of DNA samples and their potential father. Thus, each infant has a known mother and three to four potential fathers, which were assigned thanks to microsatellite analysis (see [38–40] and electronic supplementary material for further details). In addition, half-sibs were present in our dataset since several females underwent several reproductive seasons and mated with different males, and because multipaternity is possible in this species [41].

The pedigree was stored in a three-column Excel file with the following information required for the statistical analysis in AsRELM-R: individual identity, mother identity, father identity. Squares with unknown paternities were left empty, as well as maternities and paternities of 'founder individuals', that correspond to the more ancient common parent of tested individuals (see electronic supplementary material, files).

## 2.3. Phenotypic traits

We used the two personality traits that were described in Zablocki-Thomas et al. [32]. Both traits present medium repeatability in this dataset [32]. We conducted all tests during the day in daylight conditions.

### 2.3.1. Emergence tests

We conducted emergence tests using a small wooden box (18 × 18 × 31 cm). We caught animals directly in their nest box between 13.00 and 17.00, identified animals and placed a single individual in the wooden box. Next, we placed the wooden box at the entrance of the home cage of the individual. We then waited at least two minutes so that the animal could habituate and calm down from the manipulation. The test consisted of opening the trap door and recording the latency for the animal to leave the box and to return to its home cage. The test lasted 5 min maximum. Individuals that never left the box within the allotted 5 min were given a score of 300 s. We conducted this test between 1 and 13 times per individual for a total of 1238 tests. Some individuals were tested only once as they died before we could test them twice. We waited at least three weeks before repeating the test with the same individual. Repeatability for this trait reached $R = 0.33 \pm 0.04$ s.e. [32].

### 2.3.2. Agitation score

We followed the protocol described in Verdolin & Harper [42] and evaluated an agitation score between one and six times per individual, for a total of 1001 tests. In brief, the test consisted of catching the animal and recording and scoring its reaction: urinating (1pt), defecating (1pt), vocalizing (1pt), struggling (2pt) and biting (3pt). According to this protocol, animals were rated from zero to eight. The rating started directly after extraction of the animal from its nest box and lasted 30 s maximum. We rated agitation during different events of the monitoring protocols including when animal keepers conducted the monthly weighing or before physical testing. Repeatability for this trait reached $R = 0.28 \pm 0.04$ s.e. [32].

### 2.3.3. Morphology

We recorded the length of the lower arm (ulna/radius) and head width with a pair of digital callipers (±0.01 mm; Mitutoyo, Kanagawa, Japan), as reported in previous studies [32]. We extracted body weight at the time of each test from the laboratory colony database.

### 2.3.4. Life-history traits

We extracted body weight at birth, body weight at three months, litter size and mother identity from the colony database. We calculated growth rate as the weight gain in grams over the first three months of life, which is the period during which most of the growth occurs in this species [43].

Complete data were not available for all individuals, which explains why the sample size for each phenotypic trait varies and is different from the total number of individuals present in the study ($N = 486$). Indeed, we could not collect all the phenotypic data at the same time, in particular, to avoid additional stress during the behavioural tests. Some individuals could not be tested for all traits as some died or were unavailable (when involved in reproduction, for example).

## 2.4. Statistical analysis

We ran linear mixed models (also called 'animal models') with AsREML-R software (v. 3.0) [44] to obtain a restricted maximum likelihood estimation of variance and covariance components, with a pedigree incorporated to quantify the additive genetic variance. We selected models based on log-likelihood comparisons.

### 2.4.1. Univariate models

We did not transform our variables except for emergence latency, which was $\log_{10}$-transformed, as it is typically done for latency data that do not present classical distributions [45]. We treated variables as Gaussian in our models. We also added +0.5 to the logarithm as some latencies were equal to 0. To assess the relative contribution of genes to the phenotype, we first assessed the heritability ($h^2$) of our phenotypic variables with univariate models. We tested, step-by-step, the fixed effects of age, body mass and sex in interaction (because of a sexual dimorphism in body size) by running the same model without the effect and by testing it against a $\chi^2$ distribution with one degree of freedom. To test for the significance of additive genetic variance, we ran the same model with the pedigree component removed and tested it against a $\chi^2$ distribution with one degree of freedom and divided them by two (note: in all our models, body weight was always removed). To estimate maternal environment, also called 'maternal effect', we added mother identity as a random factor. To account for pseudo-replication due to repeated measurements per individuals, we added individual identity as random factor in the models [27]. We calculated the total phenotypic variance as the sum of the variance of all random components [13]. In our models, the phenotypic variance ($V_p$) is divided in three to four parameters, depending on whether there are several measurements or not

$$V_p = V_a + V_m + V_{ce} + V_{pe} + \varepsilon,$$

where $V_a$ is the additive genetic variance, $V_m$ is the variance explained by the identity of the mother, $V_{ce}$ is the common environmental variance explained by the animal's housing environment (within shared environment consistency), $V_{pe}$ is the permanent environmental variance explained by the identity of the individual (within individual consistency) [27,46].

### 2.4.2. Trait comparisons

We report the amount of genetic variance relative to the trait mean ($I_a$) [25,47,48]:

$$I_a = \frac{V_a}{\bar{X}^2}.$$

### 2.4.3. Bivariate models

We scaled all variables using the 'scale' function in R. We tested for genetic correlation between emergence latency and agitation, emergence latency and radius length, emergence latency and birth weight, emergence latency and growth rate, and genetic correlation between agitation and radius length, agitation and birth weight, agitation and growth rate (see Results, table 2). We were unable to run bivariate models with head width due to convergence issues. We then tested for the significance of two fixed effects, sex and age, by comparing likelihood ratios with and without the effect, with one degree of freedom, first with sex and then with age. We next tested for the significance of the mother effect as a random parameter by comparing the general model with a model in which the covariance

due to the maternal effect ($COV_m$) is null [46], with a likelihood ratio test with one degree of freedom. When the mother effect was not significant or caused convergence issues, we removed it from the model. We tested for the significance of covariance due to the additive effect of genes ($COV_a$) as previously described by comparing the general model with a model in which $COV_a$ is null [46].

# 3. Results

We found significant additive genetic variances for radius length, emergence latency and our agitation score (table 1). For personality, we found no significant amount of variance explained by maternal effects, but we found significant maternal effects for birth weight, growth rate, radius length and head width. The variance explained by the common environment was significant for the agitation score, birth weight, growth rate and radius length.

We found a significant and negative additive genetic covariance of emergence latency with both radius length and growth rate. We also detected a significant and positive covariance of emergence latency with growth rate when we took maternal effects into account (table 2). This suggests that mothers that produce babies with higher birth weight will also produce babies with longer emergence latencies [46]. Finally, when we estimated additive genetic covariance between morphological and birth parameters, we only detected a significant and positive additive genetic covariance between radius length and head width, and between radius length and growth rate (electronic supplementary material). The genetic correlation between head width and radius length was extremely high (0.73) and is possibly due to the allometric effect of overall body size.

# 4. Discussion

## 4.1. Trait variance decomposition

In this study, we decomposed the sources of variation of personality, morphological and life-history traits in order to compare their narrow-sense heritability. Personality traits showed significant additive genetic variance, but in contrast, only one morphological trait and none of the life-history traits showed significant additive genetic variance. As predicted, narrow-sense heritability ($h^2 = V_a / V_p$) was higher for one morphological trait compared with behavioural traits, but this was not the case for the other morphological trait. The agitation score during handling has previously been suggested to be associated with shyness and anxiety [32,42], emergence latency on the other hand has been suggested to be linked to exploration [32]. In our study, we found that both traits showed significant additive genetic variance and medium heritability (0.19–0.22). Moreover, these traits showed an amount of genetic variance relative to the trait mean that was ten times greater ($I_a \times 10^2 = 4.1$–7.3) than that of other traits. Similar to a recent study in squirrels (*Tamias striatus*) [49], we found that the permanent environment explained a moderate proportion of the variability, albeit not significantly so in our study. There was also an effect of age for the agitation score, but the effect of sex was not significant.

Consistent with the literature, heritability estimates for life-history traits (growth rate and birth weight) were relatively low [35], when compared with the other traits. Birth weight and growth rate both showed non-significant additive genetic variance and low heritability, as has been documented for other species [50,51]. Growth rate and birth weight of deer, for example, presented a similar low heritability ($h^2 = 0.11$ for males and 0.25 for females) [52]. The amount of genetic variance relative to the trait mean was ten times higher ($I_a \times 10^2 = 0.35$–0.47) for these traits compared with that for morphological traits ($I_a \times 10^2 = 0.013$–0.047).

We also detected significant maternal effects for life-history and morphological traits. Maternal effects have been shown to be important in primates such as macaques (*Macaca mulatta*) [53], but also in other taxa like bird species [54]. The impact of the mother can be either genetic or environmental. Indeed, maternal effects may be related to the maternal investment in reproduction and can, for example, be caused by egg quality in birds or the milk composition in mammals [27], both of which depend on the mother's genotype and her environment.

We also accounted for fixed effects including sex, age and body weight [55,56] in our models as they were previously described as important determinants of personality in this species [57,58]. Doing so decreased the additive genetic variance when compared with models without these effects, but overall this did not affect our conclusions. It is important to note that body weight was never retained as a fixed term in our models. Indeed, neither the interaction between sex and body weight nor body

**Table 1.** Summary of the univariate analysis of heritability on phenotypic traits calculated with the AsREML-R animal model. Significant fixed effects are reported in the first column for each trait. $h^2 = V_a/V_p$; $I_a = V_a/\bar{X}^2$. $N$ is the sample size for each model. Bold values represent significant correlations among variables ($p < \alpha = 0.05$).

| | emergence latency (log(s + 0.5)) | agitation score | birth weight (g) | growth rate (g d$^{-1}$) | radius length (mm) | head width (mm) |
|---|---|---|---|---|---|---|
| $N$ | 373 | 374 | 454 | 454 | 417 | 416 |
| fixed effect | 1 | 1 + AGE | 1 | 1 + SEX | 1 + SEX + AGE | 1 + SEX + AGE |
| $\bar{X} \pm$ SD | 3.56 ± 1.52 | 3.29 ± 1.66 | 6.56 ± 1.27 | 0.54 ± 0.095 | 28.71 ± 1.21 | 21.22 ± 0.95 |
| $V_a \pm$ s.e. | **0.515 ± 0.215** | **0.787 ± 0.352** | 0.204 ± 0.186 | $1.01 \times 10^{-3} \pm 8.93 \times 10^{-4}$ | **0.385 ± 0.173** | 0.0611 ± 0.0894 |
| | **$p = 0.005$** | **$p = 0.01$** | $p = 0.16$ | $p = 0.09$ | **$p = 0.015$** | $p = 0.38$ |
| $V_{permanent\ environment}$ (ID) ± s.e. | 0.229 ± 0.189 | 0.350 ± 0.318 | — | — | — | — |
| | $p = 0.12$ | $p = 0.13$ | | | | |
| $V_{mother} \pm$ s.e. | 0.105 ± 0.0926 | 0.0319 ± 0.166 | **0.482 ± 0.121** | **$2.41 \times 10^{-3} \pm 6.80 \times 10^{-4}$** | **0.215 ± 0.0843** | **0.101 ± 0.0489** |
| | $p = 0.11$ | $p = 0.44$ | **$p = 3.7 \times 10^{-7}$** | **$p = 1.4 \times 10^{-5}$** | **$p = 0.0013$** | **$p = 0.008$** |
| $V_{common\ environment}$ (LOT) ± s.e. | 0.0114 ± 0.0284 | **0.205 ± 0.166** | **0.0813 ± 0.0702** | **$3.601 \times 10^{-4} \pm 3.30 \times 10^{-4}$** | **0.0707 ± 0.0615** | 0.0119 ± 0.0156 |
| | $p = 0.30$ | **$p = 0.002$** | **$p = 0.005$** | **$p = 0.01$** | **$p = 0.006$** | $p = 0.11$ |
| $V_r \pm$ s.e. | 1.445 ± 0.0723 | 2.766 ± 0.158 | 0.870 ± 0.159 | $5.052 \times 10^{-3} \pm 8.6 \times 10^{-4}$ | 0.625 ± 0.147 | 0.506 ± 0.0838 |
| $h^2$ | 0.22 | 0.19 | 0.12 | 0.11 | 0.30 | 0.09 |
| $I_a \times 10^2$ | 4.1 | 7.3 | 0.47 | 0.35 | 0.047 | 0.013 |

**Table 2.** Summary of the covariance between personality and other phenotypic traits. $COV_a$, additive genetic covariance; s.e., standard error; $p$, $p$-value; $R_a$, genetic correlation; $COV_m$, covariance due to the maternal effect; $COV_r$, residual covariance. Bold values represent significant correlations among variables ($p < \alpha = 0.05$).

|  | agitation score | emergence latency (log (s) + 0.5) |  |
|---|---|---|---|
| emergence latency | fixed effect: age<br>$COV_a \pm$ s.e.: $-0.13 \pm 0.094$<br>$p = 0.17$<br>$R_a = -0.34$<br>$COV_r \pm$ s.e.: $0.12 \pm 0.082$ | — | — |
| radius length | fixed effect: age<br>$COV_a \pm$ s.e.: $0.06 \pm 0.087$<br>$p = 0.50$<br>$R_a = 0.14$<br>$COV_r \pm$ s.e.: $-0.041 \pm 0.074$ | fixed effect: age<br>$COV_a \pm$ s.e.: **$-0.20 \pm 0.08$**<br>**$p = 0.008$**<br>**$R_a = -0.87$**<br>$COV_r \pm$ s.e.: $0.14 \pm 0.083$ | — |
| birth weight | fixed effect: 1<br>$COV_a \pm$ s.e.: $-0.045 \pm 0.10$<br>$p = 0.65$<br>$R_a = -0.11$<br>$COV_r \pm$ s.e.: $-0.007 \pm 0.050$ | fixed effect: 1<br>$COV_a \pm$ s.e.: $0.014 \pm 0.10$<br>$p = 0.89$<br>$R_a = 0.05$<br>$COV_r \pm$ s.e.: $0.005 \pm 0.091$ | fixed effect: 1<br>$COV_m \pm$ s.e.: **$0.13 \pm 0.061$**<br>**$p = 0.04$**<br>**$R_m = 0.78$** |
| growth rate | fixed effect: sex + age<br>$COV_a \pm$ s.e.: $0.046 \pm 0.087$<br>$p = 0.57$<br>$R_a = 0.13$<br>$COV_r \pm$ s.e.: $-0.092 \pm 0.079$ | fixed effect: sex<br>$COV_a \pm$ s.e.: **$-0.20 \pm 0.082$**<br>**$p = 0.008$**<br>**$R_a = -0.81$**<br>$COV_r \pm$ s.e.: $0.10 \pm 0.082$ | — |

weight by itself improved the models when sex was kept. This is probably due to the sexual dimorphism in this species with females being heavier than males.

## 4.2. Genetic covariance between traits

We detected a genetic correlation between one personality trait, emergence latency and one morphological trait, radius length. This personality trait was also correlated with the two life-history traits; birth weight through the covariance with respect to the mother and with growth rate through additive genetic covariance. Previous studies have detected genetic links between personality traits and other traits, as for example in zebra fish [12], where sociability and boldness were correlated to each other and to body pigmentation and body size, with a negative correlation between body mass and risk-taking behaviour. We found few significant maternal effects except between emergence latency and birth weight. Including this additional random factor in the models often led to convergence problems. For that reason, we have presented results without the maternal effects as is done in other studies [46,50].

The genetic correlation between personality and morphology and life history is in accordance with the phenotypic correlations previously reported for this population of mouse lemurs where individuals with lower birth weight had a shorter emergence latency [32]. Moreover, adult body size and birth weight were positively correlated with the latency to start exploration in an open field test [15]. This is also in accordance with the prediction that personality should be linked to life history [59]. As such, the results of this study are in accordance with the pace-of-life syndrome hypothesis, which posits that trait variation in life-history and phenotypic traits are constrained by the environment to different combination of traits, as a result of physiological (mainly endocrine) influences [3]. In our population, a combination of low birth weight with short emergence latencies/high birth weight with long emergence latencies appeared through genetic correlations. In addition, the results of the present study suggest that there is also an important role of maternal effects in this correlation, with mothers that produced lighter babies being also the ones that produce offspring that will have lower emergence latencies.

However, we found no significant genetic correlation between the two personality traits in contrast to the predictions of the behavioural syndrome hypothesis [60]. Our results show a trend only. In other studies, correlations between exploration and docility in wild chipmunks (*Tamias striatus*) [61,62], between activity and aggression in red squirrels (*Tamiasciurus hudsonicus*) [36], and high aggression and boldness [60] have been documented. On the other hand, previous phenotypic correlation tests between these traits in our colony of mouse lemurs were also not significant [32], consistent with the theory that genetic correlations drive phenotypic correlations [63]. Genetic correlations between morphological traits in our study were generally positive and close to one as in a previous study [64] comparing six morphometric traits.

We faced major difficulties with the convergence of the bivariate models. We could have obtained better estimates of maternal variance by a pedigree with more known paternities. A more resolved pedigree would also have resulted in an improvement of the precision of our models, but we do not expect changes to the principal patterns observed. Indeed, the 'animal model' approach was developed to conduct heritability estimations with missing paternities [27], and for some model species, heritability estimations have been conducted without known paternity [51]. Non-assigned paternity can cause an underestimated additive genetic variance, however [65].

## 4.3. Effects of captivity

We deliberately chose to study a captive population of primate species of interest to several fields of research, including personality research. It is important to acknowledge the potential influence of founder effects when dealing with genetic studies on captive populations. In this species, wild populations present a relatively high level of homozygosity, which can be interpreted by the presence of small breeding units, with breeding distances that are shorter than foraging distances [66]. This population structure is also symptomatic of the forest fragmentation that started in Madagascar since its colonization by humans 2000 years ago, and that can be traced back through genetic studies to 500 years ago [67,68]. Given the low heterozygosity and small population sizes of wild populations of *M. murinus*, our results are probably transposable to wild populations.

However, heritability is subject to variation across time and populations. In addition, heritability differences could arise between wild and captive populations [69,70]. Indeed, we can assume the sources of variation, and especially environmental variation, differ significantly in the wild when compared with the laboratory [71]. Comparing captive populations of grey mouse lemurs with one another, or comparing these results with data for wild populations would be of interest. This would allow us to test whether wild animals are more constrained than captive animals and show lower additive genetic variance or if our captive population presents similar additive genetic effects for personality and life-history traits than a wild population. However, it is generally assumed that a captive population could show lower additive genetic variability than wild animals because of founder effects, but larger phenotypic variability due to reduced selection pressures [72]. It would be interesting to further explore these trait correlations and determine their potential for fitness with a long-term study on this short-lived primate in both the wild and in captivity.

## 5. Conclusion

This study provides the first evidence of additive genetic variance of two personality traits in the grey mouse lemur. It also shows how variable the sources of variation are for different types of phenotypic traits. This study also brings the first evidence of a genetic correlation between a morphological trait and a personality trait in this small primate, and highlights the importance of maternal effects. Investigating the genes underlying these correlations and the selection operating on them would be of interest to better understand the evolution of phenotypic diversity in this primate model.

Ethics. All observations and data collected respected national and international guidelines for animal welfare. All subjects included in the study were born and reared in captivity. All experiments were approved by and in accordance with the guidelines of the local institutional ethics committee. Agreement number: F91-114-1.
Data accessibility. Pedigree and phenotypic data are available within the Dryad Digital Repository: https://doi.org/10.5061/dryad.ksn02v703 [73].
Authors' contributions. P.B.Z.-T. participated in the conception and design, acquisition of data, analysis and interpretation of data. She also wrote the first draft of the manuscript. F.A. was involved in conception and design and revising of the manuscript. C.J.K. participated in the acquisition of data (mainly genetics), analysis of data and revising of the

manuscript. A.H. participated in the conception and design, acquisition of data, analysis and interpretation of data and was involved in drafting and revising the manuscript. E.P. was involved in the conception and design, acquisition of data, revising the manuscript. All authors gave final approval for publication.

Competing interests. The authors have no competing interests to declare.

Funding. This work was supported by a doctoral grant (P.B.Z.-T.) of the ENS of Lyon, Département de Biologie. Microsatellite analysis was financed by Mark Krasnow's laboratory.

Acknowledgements. We thank Éric Gueton-Estrade, Sandrine Gondor-Bazin and Lauriane Dezaire for their help with the care of the animals. We thank Justine Mezier and Grégoire Boulinguez-Ambroise for their contribution to the data collection. We thank Mark Krasnow for his technical and financial support. We thank Valérie Ducret for kindly accepting to revise the manuscript.

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
