## [Reviewer comments · Royal Society Open Science]

Review History

RSOS-190632.R0 (Original submission)

Review form: Reviewer 1

Is the manuscript scientifically sound in its present form?

Yes

Are the interpretations and conclusions justified by the results?

Yes

Is the language acceptable?

No

Is it clear how to access all supporting data?

Yes

Do you have any ethical concerns with this paper?

No

Have you any concerns about statistical analyses in this paper?

No

Recommendation?

Major revision is needed (please make suggestions in comments)

Comments to the Author(s)

This manuscript presents the quantitative genetic study of various traits, behavioral and others, in a captive population of grey mouse lemur. The experiment and the statistical analysis both seem robust. I think it would be useful to have these estimates of quantitative genetic parameters for original traits in an original species published. However the writing is sometimes clumsy, and the presentation and the discussion of the results appears inaccurate. In particular, some of the concepts of quantitative genetics are used inappropriately. In my opinion there is a fair bit of careful re-writing to do, but it can be done. Below are some comments and references to important papers to help the authors in that process.

1. Transmission vs. variation.

I do not think your study is about how traits are transmitted. Your animal models only quantify the proportion of trait variation that is transmitted (additive genetic variance) versus the proportion that is not (residual variation, but also permanent environment, common environment and maternal variation), but do not disentangle different types of transmissions. You measure "how much", not "how". Therefore I think it would be more accurate to rephrase several sentences in term of decomposing sources of variation. (in some places, the use of the word "transmission" is appropriate though)

Examples where changes could be made:

L.7

L.8

L.16

L.20

L.186

L.187

L.318

1b: A different point, that may explain the choice of the initial phrasing. The mother variance component in your animal models does NOT measure transmission. This random effect captures the similarity between individuals sharing the same mother (while accounting for additive genetic kinship and other effects); NOT the similarity with their mother. Maternal variance may or may not arise through a vertical transmission mechanism; you would need different models to know (for instance, models with trait-based maternal effects; or possibly indirect genetic effect models; cf. Chap 6 in Quantitative genetic in the wild. 2014. Charmantier, Garant, Kruuk.).

2. Selection vs. inheritance

Darwin, Fisher, the modern synthesis and modern evolutionary theory all distinguish natural selection and inheritance as two independent processes that each can occur in the absence of the other one. Selection does not imply inheritance, nor the other way around. It is important to keep this point in mind to develop cogent and consistent evolutionary arguments.

E.g.:

L.40-42: "they are subject to selection, and may have fitness consequences" is a tautology ; more importantly, selection is unrelated to inheritance. The sentence "As such, personality may be transmitted across generations" does not follow from the previous one.

L.46-48 "traits can be selected together [...] correlated through selection" I have not read this paper but the phrasing is strange, probably vague. Does "correlated through selection" mean that phenotypic correlations are created within generations by selection (or similarly that there was correlated selection involving both traits), or that genetic correlations have emerged through past selection? Or something else?

The confusion is also found in the improper use of "opportunity for selection" and "evolvability"

3. Evolvability, coefficient of variation, opportunity for selection

These terms are used in ways that I think are not standard for quantitative genetics. I am not pretending that I have the truth about the meaning of words, but I recommend the authors should consider how the words are used by other researchers and how their text will likely be interpreted.

Evolvability has indeed been defined as the capability to generate heritable and selectable variation; but confusingly, the term is not exactly used with that meaning in quantitative genetics. Instead, it is the mean standardized additive genetic variance ($e_{\mu} = I_a = V_a / \text{Mean}^2$, cf. Hansen and Houle 2008; Hansen, Pelabon and Houle 2011), and indicate how much the mean could change over time as a result of evolution given the current additive genetic variance. This definition does not consider the generation of new genetic variation, nor the presence of selection; but only the current standing genetic variation (which may or may not vary over generations).

The opportunity for selection is strictly defined as the phenotypic variance in relative fitness. If the trait you consider was fitness, " I_a " would be the additive genetic variance in relative fitness, that is, the rate of adaptive evolution by Fisher's fundamental theorem of natural selection (for instance see Queller 2017, Fundamental Theorems of Evolution; and Moorad and Wade 2013, Selection Gradients, the Opportunity for Selection, and the Coefficient of Determination). However, you do not work with fitness in this study, and cannot access these two parameters. Here, I_a is the mean standardized evolvability as defined by Hansen and Houle.

A coefficient of variation is a standard deviation divided by a mean, or the square-root of a variance, divided by the mean. That is, $\text{sqrt}(V)/M$. Not $\text{sqrt}(V/M)$. In Hansen, Pelabon and Houle (2011) Heritability is not Evolvability, it is stated, "the coefficient of additive genetic variance, CV_A , which was the measure of evolvability emphasized by Houle (1992). So it may not be very useful to present coefficients of variation here. The mean-standardized evolvability (I_a) is just the square of the coefficient of variation; one can be calculated from the other; but I_a is now preferred (again, see Hansen, Pelabon and Houle (2011)).

E.g.:

L.71 In this context "evolvability" is not the capacity to generate heritable and selectable variation. It is the amount of genetic variance relative to the trait mean (it is the current state, not a capacity for future variation)

L.150-151 " I " is the opportunity for selection only if the focal trait is fitness.

L.152 and L.181 The definition of CV_a is non-standard.

L.193 Evolvability does not mean that in this context.

L.221 In this context, (and to some extent in general too) evolvability does not increase the likelihood that a trait evolves through natural selection. A higher evolvability simply means that the trait could evolve more relative to its mean (assuming the additive genetic variance keeps being expressed in the same way). Evolvability does not make natural selection more likely,

neither here nor in general; although natural selection could change the mean of a highly evolvable trait more, that is also true of genetic drift.

L.315-316 This analysis cannot possibly estimate the opportunity for selection since it does not consider fitness.

Additional points:

L.19 "showed an impact of" suggest rephrase as "was influenced by"

L.28 "suggesting that genetic trait correlations exist". That looks like a strange thing to write. Has any researcher ever questioned the existence of genetic correlations? (certainly people have wondered about their strength, direction, evolutionary significance...)

L. 36-37 The phrasing makes the definition ambiguous. Personality is defined as consistent differences among individuals (NOT consistent variability). Personality does NOT imply that individuals have a *fixed* behavior. The behavior of a given individual may vary in the presence of personality, but individuals will consistently be different *on average*; or if you prefer, individuals could be considered to have fixed *propensities* for the given behavior.

L.45 I do not know what this sentence means. What is genetic mediation? Does that mean genetic correlation? Pleiotropy? Linkage?

L.60 "pedigree" are not strictly necessary. Suggest replacing with "kinship" or "relatedness".

L.89-93 Mention how maternities were inferred.

L.118 What database?

L.129-Table 2. Is the log-transformation done using the equation in table 2? If yes, where does the 0.5 come from? (that seems arbitrary) Why not use a generalized linear model?

L.130 I know that this is a very common misconception among biologists (in part due to a few textbooks written by biologists who were not statisticians), but there is no (and there never was) assumption of normality for the data (response or predictor) in linear (mixed) model. The normality assumption is about the residuals (or errors), and about the random effect estimates in the case of mixed models. Standard errors and p-values are computed assuming normality AFTER accounting for predictors (otherwise it would be impossible to analyze sexually-dimorphic traits, or any response variable with a strong effect of a categorical predictor...). Just check Wikipedia or any statistics textbook if you do not know about the assumptions of linear models.

L.136 Chi-square tests on variance components are conservative since variances cannot be negative. It is quite common to divide p-values of such tests by 2 to approximate correct p-values (ideally you need to fit a test with a mixture of chi-squares with 1 and 0 df; but that does not matter much). See Self and Liang 1987 Asymptotic Properties of Maximum Likelihood Estimators and Likelihood Ratio Tests Under Nonstandard Conditions; or Pinheiro and Bates book "Mixed-Effects Models in S and S-PLUS" chapter 2.

L.138-139 Something is strange in the definition of permanent environment and common environment variances. Permanent environment effects are generally defined as within individual consistency, while common environment effects are shared among individuals. I do not get that difference from your phrasing.

L.172 I do not understand this sentence. Do you mean that you found a positive covariance at the level of mothers?

Table 2. Please define COVa, SE, P and Ra in caption.

L.205 Where was body weight accounted for? I do not see that in the results. Anyways, maybe you should not account for weight: if weight is genetically correlated with other traits you consider, then including weight as a predictor would change the interpretation of quantitative genetic parameters for the focal trait (making them conditional on weight genetic variation).

L.258 "General theory" of what?

L.267-268 It sounds like you consider mother-level covariance as a component of genetic correlation. It is not. If both mother id and a pedigree are in the model, the mother variance corresponds only to the similarity between siblings independently and on top of direct additive genetic effects (although it is possible that the maternal effects have themselves an indirect additive genetic component, but that your models are agnostic about that).

L.286 "no significant genetic correlation"?

L.308-309 Maybe that is true, but it is not completely trivial to me at the moment. How exactly does inbreeding decrease additive genetic variance in your system? In particular, what type of inbreeding are you talking about: non-random mating through mate choice favoring relatives? Pedigree inbreeding? Small sample size and population structure? Could you provide references?

SI:

I am quite amazed that pedigree reconstruction seems to be done without the use of specialized software. In my experience this is difficult and very painful. It does not mean it is wrong (although it would be nice to see an evaluation of the performance of the current procedure), but I believe it would be much easier, powerful, repeatable, and reliable to use software/R-packages like COLONY, CERVUS, MasterBayes, Sequoia... I encourage the authors to explore these options for future studies.

Review form: Reviewer 2

Is the manuscript scientifically sound in its present form?

Yes

Are the interpretations and conclusions justified by the results?

Yes

Is the language acceptable?

Yes

Is it clear how to access all supporting data?

Yes

Do you have any ethical concerns with this paper?

No

Have you any concerns about statistical analyses in this paper?

No

Recommendation?

Reject

Comments to the Author(s)

Review MS RSOS-190623

In this manuscript the authors present new empirical data on heritability and genetic covariation of behavioural, life-history and morphological traits of a small primate in a captive colony. They found moderate heritability for all traits and genetic correlations between a behavioural variable related to boldness and growth rate and some morphological traits. The study uses state-of-the-art statistical approaches to analyse these data and is based on a sufficiently large data set to run these data-hungry models. In my view the study provides interesting patterns but in its current presentation is weak in integrating them into the body of current theory (e.g., pace-of-life syndromes, life-history trade-offs and behaviour etc.). Thus, it is very descriptive. Below I made some general suggestions and specific remarks to might help to improve the presentation of the study.

General comments

(1) The presentation of the study is very taxon-centered. There are good examples of heritability and genetic correlations for behavioural and life-history traits in other mammals. Just doing it (again) for a primate species, does not in itself justifies a study. Given a large body of research on life-history and behaviour of this species (also in the field), it is a very suitable study system, but this needs to be clearer in the presentation.

(2) The study is based on data from a captive colony. Captive populations are often built from a small founder population, which might cause some inbreeding effects, captivity effects, unintended directional selection for some traits etc. In the current presentation of their data the authors do not discuss any of these issues inherent of captive populations and I would urge them to do so in the discussion.

(3) This is my main concern. The authors collected a wealth of interesting data and present their findings clearly. However, their results could be discussed more directly in the light of current theory on the evolution and maintenance of (co)variation in behavioural and life-history traits. For example, the pace-of-life syndrome hypothesis makes very clear directed predictions about genetic correlations between traits. Also the life-history trade-off hypothesis lends itself to frame the study.

(4) It remains unclear how personality was quantified. The behavioural variables were quantified repeatedly but were they repeatable? What personality score did you use in the models (average?, predicted individual intercepts?)? The definition of personality is not correct.

(4) The English would need some editing.

Specific comments

Abstract

L8: See above, the aim is not well justified. Just doing a study in a primate does not justify it.

L20: Replace “in” with “on”.

Introduction

Overall, I think the introduction could develop a more theoretical framework to develop the study aims (see above). Predictions and hypotheses are clear but very descriptive and not developed from theory.

L36: What is consistent variability? Personality is not defined as consistent variability but as consistent among-individual differences. Also, personality does not imply that behaviours are “fixed”. I would urge the authors to go back to classical papers on personality (e.g., Dingemanse et al. 2010 TREE).

L41: These are incorrect references for the statement. Please cite meta-analyses on fitness consequences, not opinion papers.

L58: This is true but a rather weak justification for the study. I would suggest to motivate the study based on theory.

Methods

Please add a section on the colony (number of generations in captivity, inbreeding, breeding scheme (who with whom?), number of founders etc.).

Behavioural tests

Please specify whether these tests were performed in nocturnal conditions (activity period of the species) or diurnal conditions. Are the behavioural variables repeatable at the population level?

Statistics

How did you treat the agitation scores in the models, they are ordinary data?

How did you treat the repeated behaviours in the models (average expression, random intercepts...)?

L143: Please add the error term to the equation.

Bivariate models:

Please express clearly in the text, which pairs of variables were combined.

Results

L168: Since you have two terms for environmental variation, be explicit here.

Discussion

As stated above, the presentation of the results of the study would gain much from linking it closer with the current theoretical framework. Your results are extremely interesting, e.g. before the pace-of-life syndrome hypothesis. I would suggest to completely overwork this discussion

going away from a list-like presentation of sources of variation in the various traits and replace it with a discussion of your results on heritability and particularly of genetic correlation between the different traits.

Moreover, as stated above, please add a discussion of the captivity problem.

References

The reference list needs serious proof-reading (species names in italic, missing page numbers, missing journal names, etc.).

Tables and Figures

Fig. 1: not needed.

Review form: Reviewer 3

Is the manuscript scientifically sound in its present form?

Yes

Are the interpretations and conclusions justified by the results?

No

Is the language acceptable?

Yes

Is it clear how to access all supporting data?

No

Do you have any ethical concerns with this paper?

No

Have you any concerns about statistical analyses in this paper?

Yes

Recommendation?

Major revision is needed (please make suggestions in comments)

Comments to the Author(s)

The authors have conducted behavioural assessments and physical trait measurements on a captive and pedigreed population of Grey Mouse Lemurs. It is unusual to see studies like this in primates and their sample sizes are reasonable for a primate study. This paper is of interest to readers but is not quite up to date in methods and discussions and is not detailed enough to be replicated.

This paper is reasonably thorough but is behind the times in the following aspects:

- the calculation of heritability with fixed effects is problematic. Please read (Wilson 2008). In short, fixed effects account for variance in the model. A better approach is that of (de Villemereuil et al. 2018) to account for fixed effects in calculating heritability. The variance accounted for by fixed effects in the model is calculated and then included in the heritability calculation.

- there are papers showing differences between captive and wild population derived repeatibilities. (Bell et al. 2009) review might be an early one but there are also blue/great tit papers on this topic (Herborn et al. 2010) and somewhere some theory on differences in repeatability between captivity and the wild (I can't remember where though!). I would have liked to have seen a more nuanced introduction and discussion that included points such as this and some of the other points raised below.
- the authors can calculate standard errors on their heritability estimates using the pin() function. This would be preferable to estimates with no uncertainty.
- the assumptions behind building their model are unusual. The authors start at the assumption of heritability and add other factors including the permanent environment effect to this, according to the methods. The repeatability includes the heritability (roughly speaking, but there are caveats to that), so heritability is a portion of the greater whole. Or, another way to think about it, the calculation is the heritability of repeatability. Therefore, they would be better to start with the assumption of a permanent environment effect and add the pedigree second.

A few other points came up as I read through the paper. In the discussion, evolvability in a captive population is a nice idea, but it is a bit of a stretch to imply a link to wild populations in the discussion. Especially since the opportunity for selection and heritability depend on population size and are population specific (sorry, I forget the name of the person who has done theory work on the opportunity for selection and population size). The authors discuss some of the interests and issues on lines 304 – 311 but it comes across as a little speculative and could have more grounding in the literature. Line 239-241 also doesn't mention that heritability and evolvability are population specific, so there is great difficulty in drawing the comparison here. It would be good if the authors are more tentative about this statement.

The first paragraph of the introduction is very interesting, though it feels slightly disjointed from the rest of the text at the moment.

Line 36: this is a naughty (read:bad) way to describe personality! Especially before a paper that uses variance partitioning to explore animal personality and heritability. Animal personality is the consistency of behaviour within an individual relative to other individuals within the same population. Or, animal personality is defined statistically as a greater proportion of behavioural variance within a population being attributable to within-individual similarity than between-individual differences. There are loads of possibilities, but the current formulation misses the subtlety that animal personality is 1) a statistical concept 2) distinct from individual differences in intra-individual variability (more similar concept than animal personality to the current phrasing the authors have used!) (Stamps et al. 2012) 3) is a relative population-based measure. There is no quantifiable 'personality trait' in an individual animal, it is only by comparison that we arrive at a concept of personality.

If the authors could do a power analysis for their pedigree, this would make their paper more robust. I think (Germain et al. 2018) has one for the Mandarte island sparrows, and there is a simple one in (Winney et al. 2018).

In general, there is a lack of detail in the methods such that I could not re-do the analysis or make a full assessment of whether they were appropriate. The lack of detail includes:

- were covariance models run with repeated measures or not? If run with repeated measures, the calculation of covariance is different and the authors should check out the appendix of (Dingemanse et al. 2012).
- it seems some subjects died during the study but this is not mentioned in the initial sample sizes lines 80-85.

-authors state they transform but not that this is to improve model residuals. Also the authors don't state their distribution, so whilst they are probably running LMM because it's asreml this needs to be stated.

-do the authors include repeated measures when they calculate evolvability?

- authors need to give repeatability for their 'personality traits' to show that they are repeatable traits.

- please supply residual covariances for multivariate models so that the covariances can be viewed in context.

- line 299 mentions inbreeding coefficients, which another population e.g. the Mandarte song sparrow population (Jane Reid's papers) takes into account during their calculations of heritability. Do the authors need to consider this here?

Confusing points:

-line 126 pedigrees look like a random factor but they aren't. Re-word for clarity.

-introduction and methods: it is confusing to introduce the 'five personality traits' in the introduction and then never link them to the traits you actually measure. There is also a lot of great work on confirming that these traits match our human-imposed labels (Carter et al. 2013). It would be good if the authors read these to get an understanding of the controversy over the categorisation of behaviours.

-in table 1, is N for the number of animals? It would be worth giving overall sample size below if it is.

- line 14: define your type of heritability. Narrow-sense is mentioned once in the discussion.

- line 41: the sentence beginning 'as such' does not follow on from the previous sentence.

- line 50: the reference used here is not the first occurrence of this type of study, read work by Peter Biro e.g. (Biro & Post 2008).

- line 59-60: speculation too strongly worded. Replace with 'one of the reasons for this could be...'

Line 109: 'grabbing' has bad connotations, choose another suitable word like catching.

-line 258: 'general theory' is what theory?

Bell, A. M., Hankison, S. J., & Laskowski, K. L. (2009). The repeatability of behaviour: a meta-analysis. *Animal Behaviour*, 77(4), 771-783. <https://doi.org/10.1016/j.anbehav.2008.12.022>

Biro, P. A., & Post, J. R. (2008). Rapid depletion of genotypes with fast growth and bold personality traits from harvested fish populations. *Proceedings of the National Academy of Sciences of the United States of America*, 105(8), 2919-2922. <https://doi.org/10.1073/pnas.0708159105>

Carter, A. J., Feeney, W. E., Marshall, H. H., Cowlshaw, G., & Heinsohn, R. (2013). Animal personality: what are behavioural ecologists measuring? *Biological Reviews*, 88(2), 465-475. <https://doi.org/10.1111/brv.12007>

de Villemereuil, P., Morrissey, M. B., Nakagawa, S., & Schielzeth, H. (2018). Fixed-effect variance and the estimation of repeatabilities and heritabilities: issues and solutions. *Journal of Evolutionary Biology*, 31(4), 621-632. <https://doi.org/10.1111/jeb.13232>

Dingemanse, N. J., Dochtermann, N. a., & Nakagawa, S. (2012). Defining behavioural syndromes and the role of 'syndrome deviation' in understanding their evolution. *Behavioral Ecology and Sociobiology*, 66(11), 1543-1548. <https://doi.org/10.1007/s00265-012-1416-2>

Germain, R. R., Wolak, M. E., Reid, J. M., & Germain, R. R. (2018). Individual repeatability and heritability of divorce in a wild population. *Biology Letters*, 14, 20180061.

Herborn, K. a., Macleod, R., Miles, W. T. S., Schofield, A. N. B., Alexander, L., & Arnold, K. E. (2010). Personality in captivity reflects personality in the wild. *Animal Behaviour*, 79(4), 835-843. <https://doi.org/10.1016/j.anbehav.2009.12.026>

Stamps, J. a., Briffa, M., & Biro, P. a. (2012). Unpredictable animals: individual differences in intraindividual variability (IIV). *Animal Behaviour*, 83(6), 1325-1334. <https://doi.org/10.1016/j.anbehav.2012.02.017>

Wilson, A. J. (2008). Why h^2 does not always equal V_A/V_P ? *Journal of Evolutionary Biology*, 21(3), 647–650. <https://doi.org/10.1111/j.1420-9101.2008.01500.x>

Winney, I. S., Schroeder, J., Nakagawa, S., Hsu, Y.-H., Simons, M. J. P., Sánchez-Tójar, A., ... Burke, T. (2018). Heritability and social brood effects on personality in juvenile and adult life-history stages in a wild passerine. *Journal of Evolutionary Biology*, 31(1), 75–87. <https://doi.org/10.1111/jeb.13197>

Decision letter (RSOS-190632.R0)

08-Aug-2019

Dear Dr Zablocki-Thomas,

The editors assigned to your paper ("Heritability and genetic correlations of personality, life history, and morphology in the grey mouse lemur (*M. murinus*).") have now received comments from reviewers.

The reviewers are relatively positive about publication. However, they raise a very large number of substantive issues and points that will need careful and considered attention and revision. The reviewers agree that statistical methods are appropriate and that new analyses or additional collection of data is not required. However, the manuscript will require major revision.

We would like you to revise your paper in accordance with the referee criticisms and suggestions which can be found below (not including confidential reports to the Editor). Please note this decision does not guarantee eventual acceptance.

Please submit a copy of your revised paper before 31-Aug-2019. Please note that the revision deadline will expire at 00.00am on this date. If we do not hear from you within this time then it will be assumed that the paper has been withdrawn. In exceptional circumstances, extensions may be possible if agreed with the Editorial Office in advance. We do not allow multiple rounds of revision so we urge you to make every effort to fully address all of the comments at this stage. If deemed necessary by the Editors, your manuscript will be sent back to one or more of the original reviewers for assessment. If the original reviewers are not available, we may invite new reviewers.

- Data accessibility

<http://datadryad.org/submit?journalID=RSOS&manu=RSOS-190632>

- Competing interests

- Authors' contributions

- Acknowledgements

- Funding statement

Kind regards,

on behalf of Professor Peter Visscher (Associate Editor) and Steve Brown (Subject Editor)
 openscience@royalsociety.org

Editorial Comments to Author:

When resubmitting your manuscript, please provide the code used to run your analysis; in-line with our data and code sharing policies: <https://royalsociety.org/journals/ethics-policies/data-sharing-mining/>

Reviewers' Comments to Author:

Reviewer: 1
 Comments to the Author(s)

This manuscript presents the quantitative genetic study of various traits, behavioral and others, in a captive population of grey mouse lemur. The experiment and the statistical analysis both seem robust. I think it would be useful to have these estimates of quantitative genetic parameters for original traits in an original species published. However the writing is sometimes clumsy, and the presentation and the discussion of the results appears inaccurate. In particular, some of the concepts of quantitative genetics are used inappropriately. In my opinion there is a fair bit of careful re-writing to do, but it can be done. Below are some comments and references to important papers to help the authors in that process.

1. Transmission vs. variation.

I do not think your study is about how traits are transmitted. Your animal models only quantify the proportion of trait variation that is transmitted (additive genetic variance) versus the proportion that is not (residual variation, but also permanent environment, common environment and maternal variation), but do not disentangle different types of transmissions. You measure "how much", not "how". Therefore I think it would be more accurate to rephrase several sentences in term of decomposing sources of variation. (in some places, the use of the word "transmission" is appropriate though)

Examples where changes could be made:

L.7
 L.8
 L.16
 L.20
 L.186
 L.187

L.318

1b: A different point, that may explain the choice of the initial phrasing. The mother variance component in your animal models does NOT measure transmission. This random effect captures the similarity between individuals sharing the same mother (while accounting for additive genetic kinship and other effects); NOT the similarity with their mother. Maternal variance may or may not arise through a vertical transmission mechanism; you would need different models to know (for instance, models with trait-based maternal effects; or possibly indirect genetic effect models; cf. Chap 6 in Quantitative genetic in the wild. 2014. Charmantier, Garant, Kruuk.).

2. Selection vs. inheritance

Darwin, Fisher, the modern synthesis and modern evolutionary theory all distinguish natural selection and inheritance as two independent processes that each can occur in the absence of the other one. Selection does not imply inheritance, nor the other way around. It is important to keep this point in mind to develop cogent and consistent evolutionary arguments.

E.g.:

L.40-42: "they are subject to selection, and may have fitness consequences" is a tautology ; more importantly, selection is unrelated to inheritance. The sentence "As such, personality may be transmitted across generations" does not follow from the previous one.

L.46-48 "traits can be selected together [...] correlated through selection" I have not read this paper but the phrasing is strange, probably vague. Does "correlated through selection" mean that phenotypic correlations are created within generations by selection (or similarly that there was correlated selection involving both traits), or that genetic correlations have emerged through past selection? Or something else?

The confusion is also found in the improper use of "opportunity for selection" and "evolvability"

3. Evolvability, coefficient of variation, opportunity for selection

These terms are used in ways that I think are not standard for quantitative genetics. I am not pretending that I have the truth about the meaning of words, but I recommend the authors should consider how the words are used by other researchers and how their text will likely be interpreted.

Evolvability has indeed been defined as the capability to generate heritable and selectable variation; but confusingly, the term is not exactly used with that meaning in quantitative genetics. Instead, it is the mean standardized additive genetic variance ($e_{\mu} = I_a = V_a / \text{Mean}^2$, cf. Hansen and Houle 2008; Hansen, Pelabon and Houle 2011), and indicate how much the mean could change over time as a result of evolution given the current additive genetic variance. This definition does not consider the generation of new genetic variation, nor the presence of selection; but only the current standing genetic variation (which may or may not vary over generations).

The opportunity for selection is strictly defined as the phenotypic variance in relative fitness. If the trait you consider was fitness, " I_a " would be the additive genetic variance in relative fitness, that is, the rate of adaptive evolution by Fisher's fundamental theorem of natural selection (for instance see Queller 2017, Fundamental Theorems of Evolution; and Moorad and Wade 2013, Selection Gradients, the Opportunity for Selection, and the Coefficient of Determination). However, you do not work with fitness in this study, and cannot access these two parameters. Here, I_a is the mean standardized evolvability as defined by Hansen and Houle.

A coefficient of variation is a standard deviation divided by a mean, or the square-root of a variance, divided by the mean. That is, \sqrt{V}/M . Not $\sqrt{V/M}$. In Hansen, Pelabon and Houle (2011) Heritability is not Evolvability, it is stated, "the coefficient of additive genetic variance, CV

A, which was the measure of evolvability emphasized by Houle (1992). So it may not be very useful to present coefficients of variation here. The mean-standardized evolvability (I_a) is just the square of the coefficient of variation; one can be calculated from the other; but I_a is now preferred (again, see Hansen, Pelabon and Houle (2011)).

E.g.:

L.71 In this context "evolvability" is not the capacity to generate heritable and selectable variation. It is the amount of genetic variance relative to the trait mean (it is the current state, not a capacity for future variation)

L.150-151 "I" is the opportunity for selection only if the focal trait is fitness.

L.152 and L.181 The definition of CVa is non-standard.

L.193 Evolvability does not mean that in this context.

L.221 In this context, (and to some extent in general too) evolvability does not increase the likelihood that a trait evolves through natural selection. A higher evolvability simply means that the trait could evolve more relative to its mean (assuming the additive genetic variance keeps being expressed in the same way). Evolvability does not make natural selection more likely, neither here nor in general; although natural selection could change the mean of a highly evolvable trait more, that is also true of genetic drift.

L.315-316 This analysis cannot possibly estimate the opportunity for selection since it does not consider fitness.

Additional points:

L.19 "showed an impact of" suggest rephrase as "was influenced by"

L.28 "suggesting that genetic trait correlations exist". That looks like a strange thing to write. Has any researcher ever questioned the existence of genetic correlations? (certainly people have wondered about their strength, direction, evolutionary significance...)

L. 36-37 The phrasing makes the definition ambiguous. Personality is defined as consistent differences among individuals (NOT consistent variability). Personality does NOT imply that individuals have a *fixed* behavior. The behavior of a given individual may vary in the presence of personality, but individuals will consistently be different *on average*; or if you prefer, individuals could be considered to have fixed *propensities* for the given behavior.

L.45 I do not know what this sentence means. What is genetic mediation? Does that mean genetic correlation? Pleiotropy? Linkage?

L.60 "pedigree" are not strictly necessary. Suggest replacing with "kinship" or "relatedness".

L.89-93 Mention how maternities were inferred.

L.118 What database?

L.129-Table 2. Is the log-transformation done using the equation in table 2? If yes, where does the 0.5 come from? (that seems arbitrary) Why not use a generalized linear model?

L.130 I know that this is a very common misconception among biologists (in part due to a few textbooks written by biologists who were not statisticians), but there is no (and there never was) assumption of normality for the data (response or predictor) in linear (mixed) model. The normality assumption is about the residuals (or errors), and about the random effect estimates in the case of mixed models. Standard errors and p-values are computed assuming normality AFTER accounting for predictors (otherwise it would be impossible to analyze sexually-

dimorphic traits, or any response variable with a strong effect of a categorical predictor...). Just check Wikipedia or any statistics textbook if you do not know about the assumptions of linear models.

L.136 Chi-square tests on variance components are conservative since variances cannot be negative. It is quite common to divide p-values of such tests by 2 to approximate correct p-values (ideally you need to fit a test with a mixture of chi-squares with 1 and 0 df; but that does not matter much). See Self and Liang 1987 *Asymptotic Properties of Maximum Likelihood Estimators and Likelihood Ratio Tests Under Nonstandard Conditions*; or Pinheiro and Bates book "Mixed-Effects Models in S and S-PLUS" chapter 2.

L.138-139 Something is strange in the definition of permanent environment and common environment variances. Permanent environment effects are generally defined as within individual consistency, while common environment effects are shared among individuals. I do not get that difference from your phrasing.

L.172 I do not understand this sentence. Do you mean that you found a positive covariance at the level of mothers?

Table 2. Please define COVa, SE, P and Ra in caption.

L.205 Where was body weight accounted for? I do not see that in the results. Anyways, maybe you should not account for weight: if weight is genetically correlated with other traits you consider, then including weight as a predictor would change the interpretation of quantitative genetic parameters for the focal trait (making them conditional on weight genetic variation).

L.258 "General theory" of what?

L.267-268 It sounds like you consider mother-level covariance as a component of genetic correlation. It is not. If both mother id and a pedigree are in the model, the mother variance corresponds only to the similarity between siblings independently and on top of direct additive genetic effects (although it is possible that the maternal effects have themselves an indirect additive genetic component, but that your models are agnostic about that).

L.286 "no significant genetic correlation"?

L.308-309 Maybe that is true, but it is not completely trivial to me at the moment. How exactly does inbreeding decrease additive genetic variance in your system? In particular, what type of inbreeding are you talking about: non-random mating through mate choice favoring relatives? Pedigree inbreeding? Small sample size and population structure? Could you provide references?

SI:

I am quite amazed that pedigree reconstruction seems to be done without the use of specialized software. In my experience this is difficult and very painful. It does not mean it is wrong (although it would be nice to see an evaluation of the performance of the current procedure), but I believe it would be much easier, powerful, repeatable, and reliable to use software/R-packages like COLONY, CERVUS, MasterBayes, Sequoia... I encourage the authors to explore these options for future studies.

Reviewer: 2
 Comments to the Author(s)

Review MS RSOS-190623

In this manuscript the authors present new empirical data on heritability and genetic covariation of behavioural, life-history and morphological traits of a small primate in a captive colony. They found moderate heritability for all traits and genetic correlations between a behavioural variable related to boldness and growth rate and some morphological traits. The study uses state-of-the-art statistical approaches to analyse these data and is based on a sufficiently large data set to run these data-hungry models. In my view the study provides interesting patterns but in its current presentation is weak in integrating them into the body of current theory (e.g., pace-of-life syndromes, life-history trade-offs and behaviour etc.). Thus, it is very descriptive. Below I made some general suggestions and specific remarks to might help to improve the presentation of the study.

General comments

- (1) The presentation of the study is very taxon-centered. There are good examples of heritability and genetic correlations for behavioural and life-history traits in other mammals. Just doing it (again) for a primate species, does not in itself justifies a study. Given a large body of research on life-history and behaviour of this species (also in the field), it is a very suitable study system, but this needs to be clearer in the presentation.
- (2) The study is based on data from a captive colony. Captive populations are often built from a small founder population, which might cause some inbreeding effects, captivity effects, unintended directional selection for some traits etc. In the current presentation of their data the authors do not discuss any of these issues inherent of captive populations and I would urge them to do so in the discussion.
- (3) This is my main concern. The authors collected a wealth of interesting data and present their findings clearly. However, their results could be discussed more directly in the light of current theory on the evolution and maintenance of (co)variation in behavioural and life-history traits. For example, the pace-of-life syndrome hypothesis makes very clear directed predictions about genetic correlations between traits. Also the life-history trade-off hypothesis lends itself to frame the study.
- (4) It remains unclear how personality was quantified. The behavioural variables were quantified repeatedly but were they repeatable? What personality score did you use in the models (average?, predicted individual intercepts?)? The definition of personality is not correct.
- (4) The English would need some editing.

Specific comments

Abstract

L8: See above, the aim is not well justified. Just doing a study in a primate does not justify it.

L20: Replace "in" with "on".

Introduction

Overall, I think the introduction could develop a more theoretical framework to develop the study aims (see above). Predictions and hypotheses are clear but very descriptive and not developed from theory.

L36: What is consistent variability? Personality is not defined as consistent variability but as consistent among-individual differences. Also, personality does not imply that behaviours are "fixed". I would urge the authors to go back to classical papers on personality (e.g., Dingemanse et al. 2010 TREE).

L41: These are incorrect references for the statement. Please cite meta-analyses on fitness consequences, not opinion papers.

L58: This is true but a rather weak justification for the study. I would suggest to motivate the study based on theory.

Methods

Please add a section on the colony (number of generations in captivity, inbreeding, breeding scheme (who with whom?), number of founders etc.).

Behavioural tests

Please specify whether these tests were performed in nocturnal conditions (activity period of the species) or diurnal conditions. Are the behavioural variables repeatable at the population level?

Statistics

How did you treat the agitation scores in the models, they are ordinary data?

How did you treat the repeated behaviours in the models (average expression, random intercepts...)?

L143: Please add the error term to the equation.

Bivariate models:

Please express clearly in the text, which pairs of variables were combined.

Results

L168: Since you have two terms for environmental variation, be explicit here.

Discussion

As stated above, the presentation of the results of the study would gain much from linking it closer with the current theoretical framework. Your results are extremely interesting, e.g. before the pace-of-life syndrome hypothesis. I would suggest to completely overwork this discussion going away from a list-like presentation of sources of variation in the various traits and replace it with a discussion of your results on heritability and particularly of genetic correlation between the different traits.

Moreover, as stated above, please add a discussion of the captivity problem.

References

The reference list needs serious proof-reading (species names in italic, missing page numbers, missing journal names, etc.).

Tables and Figures

Fig. 1: not needed.

Reviewer: 3

Comments to the Author(s)

The authors have conducted behavioural assessments and physical trait measurements on a captive and pedigreed population of Grey Mouse Lemurs. It is unusual to see studies like this in primates and their sample sizes are reasonable for a primate study. This paper is of interest to readers but is not quite up to date in methods and discussions and is not detailed enough to be replicated.

This paper is reasonably thorough but is behind the times in the following aspects:

- the calculation of heritability with fixed effects is problematic. Please read (Wilson 2008). In short, fixed effects account for variance in the model. A better approach is that of (de Villemereuil et al. 2018) to account for fixed effects in calculating heritability. The variance accounted for by fixed effects in the model is calculated and then included in the heritability calculation.
- there are papers showing differences between captive and wild population derived repeatibilities. (Bell et al. 2009) review might be an early one but there are also blue/great tit

papers on this topic (Herborn et al. 2010) and somewhere some theory on differences in repeatability between captivity and the wild (I can't remember where though!). I would have liked to have seen a more nuanced introduction and discussion that included points such as this and some of the other points raised below.

- the authors can calculate standard errors on their heritability estimates using the pin() function. This would be preferable to estimates with no uncertainty.
- the assumptions behind building their model are unusual. The authors start at the assumption of heritability and add other factors including the permanent environment effect to this, according to the methods. The repeatability includes the heritability (roughly speaking, but there are caveats to that), so heritability is a portion of the greater whole. Or, another way to think about it, the calculation is the heritability of repeatability. Therefore, they would be better to start with the assumption of a permanent environment effect and add the pedigree second.

A few other points came up as I read through the paper. In the discussion, evolvability in a captive population is a nice idea, but it is a bit of a stretch to imply a link to wild populations in the discussion. Especially since the opportunity for selection and heritability depend on population size and are population specific (sorry, I forget the name of the person who has done theory work on the opportunity for selection and population size). The authors discuss some of the interests and issues on lines 304 – 311 but it comes across as a little speculative and could have more grounding in the literature. Line 239-241 also doesn't mention that heritability and evolvability are population specific, so there is great difficulty in drawing the comparison here. It would be good if the authors are more tentative about this statement.

The first paragraph of the introduction is very interesting, though it feels slightly disjointed from the rest of the text at the moment.

Line 36: this is a naughty (read:bad) way to describe personality! Especially before a paper that uses variance partitioning to explore animal personality and heritability. Animal personality is the consistency of behaviour within an individual relative to other individuals within the same population. Or, animal personality is defined statistically as a greater proportion of behavioural variance within a population being attributable to within-individual similarity than between-individual differences. There are loads of possibilities, but the current formulation misses the subtlety that animal personality is 1) a statistical concept 2) distinct from individual differences in intra-individual variability (more similar concept than animal personality to the current phrasing the authors have used!) (Stamps et al. 2012) 3) is a relative population-based measure. There is no quantifiable 'personality trait' in an individual animal, it is only by comparison that we arrive at a concept of personality.

If the authors could do a power analysis for their pedigree, this would make their paper more robust. I think (Germain et al. 2018) has one for the Mandarte island sparrows, and there is a simple one in (Winney et al. 2018).

In general, there is a lack of detail in the methods such that I could not re-do the analysis or make a full assessment of whether they were appropriate. The lack of detail includes:

- were covariance models run with repeated measures or not? If run with repeated measures, the calculation of covariance is different and the authors should check out the appendix of (Dingemans et al. 2012).
- it seems some subjects died during the study but this is not mentioned in the initial sample sizes lines 80-85.
- authors state they transform but not that this is to improve model residuals. Also the authors don't state their distribution, so whilst they are probably running LMM because it's asreml this needs to be stated.
- do the authors include repeated measures when they calculate evolvability?

- authors need to give repeatability for their 'personality traits' to show that they are repeatable traits.
- please supply residual covariances for multivariate models so that the covariances can be viewed in context.
- line 299 mentions inbreeding coefficients, which another population e.g. the Mandarte song sparrow population (Jane Reid's papers) takes into account during their calculations of heritability. Do the authors need to consider this here?

Confusing points:

- line 126 pedigrees look like a random factor but they aren't. Re-word for clarity.
- introduction and methods: it is confusing to introduce the 'five personality traits' in the introduction and then never link them to the traits you actually measure. There is also a lot of great work on confirming that these traits match our human-imposed labels (Carter et al. 2013). It would be good if the authors read these to get an understanding of the controversy over the categorisation of behaviours.
- in table 1, is N for the number of animals? It would be worth giving overall sample size below if it is.
- line 14: define your type of heritability. Narrow-sense is mentioned once in the discussion.
- line 41: the sentence beginning 'as such' does not follow on from the previous sentence.
- line 50: the reference used here is not the first occurrence of this type of study, read work by Peter Biro e.g. (Biro & Post 2008).
- line 59-60: speculation too strongly worded. Replace with 'one of the reasons for this could be...'
- Line 109: 'grabbing' has bad connotations, choose another suitable word like catching.
- line 258: 'general theory' is what theory?

- Bell, A. M., Hankison, S. J., & Laskowski, K. L. (2009). The repeatability of behaviour: a meta-analysis. *Animal Behaviour*, 77(4), 771–783. <https://doi.org/10.1016/j.anbehav.2008.12.022>
- Biro, P. A., & Post, J. R. (2008). Rapid depletion of genotypes with fast growth and bold personality traits from harvested fish populations. *Proceedings of the National Academy of Sciences of the United States of America*, 105(8), 2919–2922. <https://doi.org/10.1073/pnas.0708159105>
- Carter, A. J., Feeney, W. E., Marshall, H. H., Cowlshaw, G., & Heinsohn, R. (2013). Animal personality: what are behavioural ecologists measuring? *Biological Reviews*, 88(2), 465–475. <https://doi.org/10.1111/brv.12007>
- de Villemereuil, P., Morrissey, M. B., Nakagawa, S., & Schielzeth, H. (2018). Fixed-effect variance and the estimation of repeatabilities and heritabilities: issues and solutions. *Journal of Evolutionary Biology*, 31(4), 621–632. <https://doi.org/10.1111/jeb.13232>
- Dingemanse, N. J., Dochtermann, N. a., & Nakagawa, S. (2012). Defining behavioural syndromes and the role of 'syndrome deviation' in understanding their evolution. *Behavioral Ecology and Sociobiology*, 66(11), 1543–1548. <https://doi.org/10.1007/s00265-012-1416-2>
- Germain, R. R., Wolak, M. E., Reid, J. M., & Germain, R. R. (2018). Individual repeatability and heritability of divorce in a wild population. *Biology Letters*, 14, 20180061.
- Herborn, K. a., Macleod, R., Miles, W. T. S., Schofield, A. N. B., Alexander, L., & Arnold, K. E. (2010). Personality in captivity reflects personality in the wild. *Animal Behaviour*, 79(4), 835–843. <https://doi.org/10.1016/j.anbehav.2009.12.026>
- Stamps, J. a., Briffa, M., & Biro, P. a. (2012). Unpredictable animals: individual differences in intraindividual variability (IIV). *Animal Behaviour*, 83(6), 1325–1334. <https://doi.org/10.1016/j.anbehav.2012.02.017>
- Wilson, A. J. (2008). Why h^2 does not always equal V_A/V_P ? *Journal of Evolutionary Biology*, 21(3), 647–650. <https://doi.org/10.1111/j.1420-9101.2008.01500.x>
- Winney, I. S., Schroeder, J., Nakagawa, S., Hsu, Y.-H., Simons, M. J. P., Sánchez-Tójar, A., ... Burke, T. (2018). Heritability and social brood effects on personality in juvenile and adult life-

history stages in a wild passerine. *Journal of Evolutionary Biology*, 31(1), 75–87.
<https://doi.org/10.1111/jeb.13197>

Author's Response to Decision Letter for (RSOS-190632.R0)

See Appendix A.

Decision letter (RSOS-190632.R1)

04-Oct-2019

Dear Dr Zablocki-Thomas,

I am pleased to inform you that your manuscript entitled "Heritability and genetic correlations of personality, life history, and morphology in the grey mouse lemur (*M. murinus*).\" is now accepted for publication in Royal Society Open Science.

on behalf of Professor Peter Visscher (Associate Editor) and Steve Brown (Subject Editor)
openscience@royalsociety.org

Appendix A

Dear Editors,

Please find enclosed a revised version of our manuscript entitled 'Heritability and genetic correlations of personality, life history, and morphology in the grey mouse lemur (*M. murinus*)' that we would like to resubmit to Royal Society Open Science. We would like to thank you for obtaining helpful and constructive comments on our paper that we have incorporated in the revised version. Please find below a point-by-point reply to the comments of the referees. We hope that our manuscript is now suitable for publication in Royal Society Open Science.

Sincerely,

The authors

Reviewers' Comments to Author:

This manuscript presents the quantitative genetic study of various traits, behavioral and others, in a captive population of grey mouse lemur. The experiment and the statistical analysis both seem robust. I think it would be useful to have these estimates of quantitative genetic parameters for original traits in an original species published. However, the writing is sometimes clumsy, and the presentation and the discussion of the results appears inaccurate. In particular, some of the concepts of quantitative genetics are used inappropriately. In my opinion there is a fair bit of careful re-writing to do, but it can be done. Below are some comments and references to important papers to help the authors in that process.

OUR REPLY: we thank the reviewer for the many helpful conceptual remarks. We have modified the paper as indicated below.

1. Transmission vs. variation.

I do not think your study is about how traits are transmitted. Your animal models only quantify the proportion of trait variation that is transmitted (additive genetic variance) versus the proportion that is not (residual variation, but also permanent environment, common environment and maternal variation), but do not disentangle different types of transmissions. You measure "how much", not "how". Therefore I think it would be more accurate to rephrase several sentences in term of decomposing sources of variation. (in some places, the use of the word "transmission" is appropriate though)

OUR REPLY: we agree with the reviewer and have made changes throughout the manuscript as suggested.

Examples where changes could be made:

L.7 and L.8

OUR REPLY: We changed the phrasing for "heritability" and "how the sources of variation in these traits are decomposed" (L. 9-10)

L.16

OUR REPLY: **“Other factors including maternal effects also influence the sources of variation in life history and morphological trait” (L. 18-20)**

L.20

OUR REPLY: **“on the partitioning of sources of variation in personality” (L. 23-24)**

L.186

OUR REPLY: **We changed the heading of this part to: “Trait variance decomposition” (L.253)**

L.187

OUR REPLY: **We changed to: “we decomposed the sources of variation” (L. 254)**

L.318

OUR REPLY: **We changed to: “additive genetic variance” (L.361)**

1b: A different point, that may explain the choice of the initial phrasing. The mother variance component in your animal models does NOT measure transmission. This random effect captures the similarity between individuals sharing the same mother (while accounting for additive genetic kinship and other effects); NOT the similarity with their mother. Maternal variance may or may not arise through a vertical transmission mechanism; you would need different models to know (for instance, models with trait-based maternal effects; or possibly indirect genetic effect models; cf. Chap 6 in Quantitative genetic in the wild. 2014. Charmantier, Garant, Kruuk.).

OUR REPLY: **we thank the reviewer for enlightening us on this point. We had not understood this properly.**

2. Selection vs. inheritance

Darwin, Fisher, the modern synthesis and modern evolutionary theory all distinguish natural selection and inheritance as two independent processes that each can occur in the absence of the other one. Selection does not imply inheritance, nor the other way around. It is important to keep this point in mind to develop cogent and consistent evolutionary arguments.

OUR REPLY: **We completely agree with the reviewer.**

E.g.:

L.40-42: "they are subject to selection, and may have fitness consequences" is a tautology ; more importantly, selection is unrelated to inheritance. The sentence "As such, personality may be transmitted across generations" does not follow from the previous one.

OUR REPLY: **This sentence was deleted.**

L.46-48 "traits can be selected together [...] correlated through selection" I have not read this

paper but the phrasing is strange, probably vague. Does "correlated through selection" mean that phenotypic correlations are created within generations by selection (or similarly that there was correlated selection involving both traits), or that genetic correlations have emerged through past selection? Or something else?

OUR REPLY: we have modified this and hope that it has become clearer. (Second paragraph of the introduction)

The confusion is also found in the improper use of "opportunity for selection" and "evolvability"

3. Evolvability, coefficient of variation, opportunity for selection

These terms are used in ways that I think are not standard for quantitative genetics. I am not pretending that I have the truth about the meaning of words, but I recommend the authors should consider how the words are used by other researchers and how their text will likely be interpreted.

OUR REPLY: we thank you for your clarification. We understand that it is particularly important to use these terms accurately to be understood correctly by the readers. "Evolvability" is indeed a source of confusion, so we replaced it at most occurrences, even if it may render the phrasing less concise.

Evolvability has indeed been defined as the capability to generate heritable and selectable variation; but confusingly, the term is not exactly used with that meaning in quantitative genetics. Instead, it is the mean standardized additive genetic variance ($e_{\mu} = I_a = V_a / \text{Mean}^2$, cf. Hansen and Houle 2008; Hansen, Pelabon and Houle 2011), and indicate how much the mean could change over time as a result of evolution given the current additive genetic variance. This definition does not consider the generation of new genetic variation, nor the presence of selection; but only the current standing genetic variation (which may or may not vary over generations).

OUR REPLY: we agree with the reviewer.

The opportunity for selection is strictly defined as the phenotypic variance in relative fitness. If the trait you consider was fitness, " I_a " would be the additive genetic variance in relative fitness, that is, the rate of adaptive evolution by Fisher's fundamental theorem of natural selection (for instance see Queller 2017, Fundamental Theorems of Evolution; and Moorad and Wade 2013, Selection Gradients, the Opportunity for Selection, and the Coefficient of Determination). However, you do not work with fitness in this study, and cannot access these two parameters. Here, I_a is the mean standardized evolvability as defined by Hansen and Houle.

OUR REPLY: we thank the reviewer for pointing this out so clearly.

A coefficient of variation is a standard deviation divided by a mean, or the square-root of a variance, divided by the mean. That is, \sqrt{V}/M . Not $\sqrt{V/M}$. In Hansen, Pelabon and Houle (2011) Heritability is not Evolvability, it is stated, "the coefficient of additive genetic variance, CV_A , which was the measure of evolvability emphasized by Houle (1992). So it may not be very useful to present coefficients of variation here. The mean-standardized

evolvability (Ia) is just the square of the coefficient of variation; one can be calculated from the other; but Ia is now preferred (again, see Hansen, Pelabon and Houle (2011)).

OUR REPLY: **As this parameter seems redundant, we deleted references to CVa to be more concise.**

E.g.:

L.71 In this context "evolvability" is not the capacity to generate heritable and selectable variation. It is the amount of genetic variance relative to the trait mean (it is the current state, not a capacity for future variation)

OUR REPLY: **We changed the phrasing: “ to estimate key variance components of morphological and personality traits and compare their heritability to better understand their evolutionary path.” (L. 95-96)**

L.150-151 "I" is the opportunity for selection only if the focal trait is fitness.

OUR REPLY: **We rephrased with: the amount of genetic variance relative to the trait mean.**

L.152 and L.181 The definition of CVa is non-standard.

OUR REPLY: **We deleted references to CVa for clarity, as it was no longer needed.**

L.193 Evolvability does not mean that in this context.

OUR REPLY: **This part was rephrased and the term “evolvability” was removed.**

L.221 In this context, (and to some extent in general too) evolvability does not increase the likelihood that a trait evolves through natural selection. A higher evolvability simply means that the trait could evolve more relative to its mean (assuming the additive genetic variance keeps being expressed in the same way). Evolvability does not make natural selection more likely, neither here nor in general; although natural selection could change the mean of a highly evolvable trait more, that is also true of genetic drift.

OUR REPLY: **Along with other modifications in the discussion, this sentence was removed.**

L.315-316 This analysis cannot possibly estimate the opportunity for selection since it does not consider fitness.

OUR REPLY: **The term “opportunity for selection” was removed.**

Additional points:

L.19 "showed an impact of" suggest rephrase as "was influenced by"

OUR REPLY: **We rephrased according to suggestion. (L. 22)**

L.28 "suggesting that genetic trait correlations exist". That looks like a strange thing to write.

Has any researcher ever questioned the existence of genetic correlations? (certainly people have wondered about their strength, direction, evolutionary significance...)

OUR REPLY: **We changed “exist” for “underlie these trait changes”. (L.36)**

L. 36-37 The phrasing makes the definition ambiguous. Personality is defined as consistent differences among individuals (NOT consistent variability). Personality does NOT imply that individuals have a *fixed* behavior. The behavior of a given individual may vary in the presence of personality, but individuals will consistently be different *on average*; or if you prefer, individuals could be considered to have fixed *propensities* for the given behavior.

OUR REPLY: **We changed the phrasing of the definition of personality. (L.60-68)**

L.45 I do not know what this sentence means. What is genetic mediation? Does that mean genetic correlation? Pleiotropy? Linkage?

OUR REPLY: **We changed “mediation” for “correlation” (L.46)**

L.60 "pedigree" are not strictly necessary. Suggest replacing with "kinship" or "relatedness".

OUR REPLY: **This sentence was deleted along with other modifications.**

L.89-93 Mention how maternities were inferred.

OUR REPLY: **We added: “Maternities are known as mothers are isolated when pregnant and raised for at least three months with their babies.” (L. 129-130)**

L.118 What database?

OUR REPLY: **The database refers to the colony database, where all information available on individuals are stored. We added the term ‘colony’ in the manuscript.**

L.129-Table 2. Is the log-transformation done using the equation in table 2? If yes, where does the 0.5 come from? (that seems arbitrary) Why not use a generalized linear model?

OUR REPLY: **Emergence latency, as many latency measurements, generally has a non-model distribution (they are not Gaussian, or Poisson...) and are often truncated. The best way we found to make an acceptable use of these data was to log-transform them. As some individuals presented emergence latency equal to zero (they immediately emerged from the box at opening), we had to add +0.5 to avoid log(0) in our models. It is also possible to add +1, but there is still a debate on which one is the best between +0.5 and +1. We added a short explanation in the manuscript. :” We did not transform our variables except for emergence latency, which was log10-transformed, as is done for latency data that do not present classic distributions (46). We also added +0.5 in the logarithm to avoid the incompatibilities with null values in emergence latencies.”**

A similar method is also used in

Winney, I. S., Schroeder, J., Nakagawa, S., Hsu, Y. H., Simons, M. J. P., Sánchez-Tójar, A., et al. (2018). Heritability and social brood effects on personality in juvenile and adult

life-history stages in a wild passerine. *Journal of Evolutionary Biology*, 31(1), 75–87.
doi:10.1111/jeb.13197

L.130 I know that this is a very common misconception among biologists (in part due to a few textbooks written by biologists who were not statisticians), but there is no (and there never was) assumption of normality for the data (response or predictor) in linear (mixed) model. The normality assumption is about the residuals (or errors), and about the random effect estimates in the case of mixed models. Standard errors and p-values are computed assuming normality AFTER accounting for predictors (otherwise it would be impossible to analyze sexually-dimorphic traits, or any response variable with a strong effect of a categorical predictor...). Just check Wikipedia or any statistics textbook if you do not know about the assumptions of linear models.

OUR REPLY: **Indeed, I think I remember I got this ‘misconception’ from university. We removed the “normality” assumption from the methods and rephrased the methods.**

L.136 Chi-square tests on variance components are conservative since variances cannot be negative. It is quite common to divide p-values of such tests by 2 to approximate correct p-values (ideally you need to fit a test with a mixture of chi-squares with 1 and 0 df; but that does not matter much). See Self and Liang 1987 Asymptotic Properties of Maximum Likelihood Estimators and Likelihood Ratio Tests Under Nonstandard Conditions; or Pinheiro and Bates book "Mixed-Effects Models in S and S-PLUS" chapter 2.

OUR REPLY: **We agree with the reviewer and have divided the P values as suggested.**

L.138-139 Something is strange in the definition of permanent environment and common environment variances. Permanent environment effects are generally defined as within individual consistency, while common environment effects are shared among individuals. I do not get that difference from your phrasing.

OUR REPLY: **Common environment is the shared environment (more precisely the sections of the colony building) and the permanent environment is indeed the identity of the individual that refers to their “own environment”.**

L.172 I do not understand this sentence. Do you mean that you found a positive covariance at the level of mothers?

OUR REPLY: **This means that mothers that produce heavier babies also produce babies with longer emergence latencies, regardless of genetics. We completed the sentence and added a reference.**

Wilson, A. J., Réale, D., Clements, M. N., Morrissey, M. M., Postma, E., Walling, C. a, et al. (2010). An ecologist’s guide to the animal model. *The Journal of animal ecology*, 79(1), 13–26. doi:10.1111/j.1365-2656.2009.01639.x

Table 2. Please define COVa, SE, P and Ra in caption.

OUR REPLY: **We added the definitions. (L.243-245)**

L.205 Where was body weight accounted for? I do not see that in the results. Anyways, maybe you should not account for weight: if weight is genetically correlated with other traits you consider, then including weight as a predictor would change the interpretation of quantitative genetic parameters for the focal trait (making them conditional on weight genetic variation).

OUR REPLY: **We specified this in the methods: body weight was added as a fixed term interacting with sex (females are heavier in this species). When adding body weight in the model did not improved the models (especially when sex was already taken into account), this parameter was not kept. In the end, body weight was never kept as a fixed term. (L.184-187)**

L.258 "General theory" of what?

OUR REPLY: **This part was rephrased.**

L.267-268 It sounds like you consider mother-level covariance as a component of genetic correlation. It is not. If both mother id and a pedigree are in the model, the mother variance corresponds only to the similarity between siblings independently and on top of direct additive genetic effects (although it is possible that the maternal effects have themselves an indirect additive genetic component, but that your models are agnostic about that).

OUR REPLY: **We deleted the word “genetic” that was confusing.**

L.286 "no significant genetic correlation"?

OUR REPLY: **We added “significant”. (L.316)**

L.308-309 Maybe that is true, but it is not completely trivial to me at the moment. How exactly does inbreeding decrease additive genetic variance in your system? In particular, what type of inbreeding are you talking about: non-random mating through mate choice favoring relatives? Pedigree inbreeding? Small sample size and population structure? Could you provide references?

OUR REPLY: **This is a mistake. We meant “founder effect”, that makes more sense here. This was corrected. (L.337)**

Pelletier, F., Réale, D., Watters, J., Boakes, E. H., & Garant, D. (2009). Value of captive populations for quantitative genetics research. *Trends in Ecology and Evolution*, 24(5), 263–270. doi:10.1016/j.tree.2008.11.013

SI:I am quite amazed that pedigree reconstruction seems to be done without the use of specialized software. In my experience this is difficult and very painful. It does not mean it is wrong (although it would be nice to see an evaluation of the performance of the current procedure), but I believe it would be much easier, powerful, repeatable, and reliable to use software/R-packages like COLONY, CERVUS, MasterBayes, Sequoia... I encourage the authors to explore these options for future studies.

OUR REPLY: Software was not needed because of the small pool of candidate fathers (2 or 3) for each offspring. Indeed, during the reproductive season, 2 or 3 females are matched with 2 or 3 males that are known. This process ensure the highest probability of pregnancy according to the breeding experience in the lab. But indeed, specific software is needed when the dataset is big and when we do not know potential fathers.

Reviewer: 2

In this manuscript the authors present new empirical data on heritability and genetic covariation of behavioural, life-history and morphological traits of a small primate in a captive colony. They found moderate heritability for all traits and genetic correlations between a behavioural variable related to boldness and growth rate and some morphological traits. The study uses state-of-the-art statistical approaches to analyse these data and is based on a sufficiently large data set to run these data-hungry models. In my view the study provides interesting patterns but in its current presentation is weak in integrating them into the body of current theory (e.g., pace-of-life syndromes, life-history trade-offs and behaviour etc.). Thus, it is very descriptive. Below I made some general suggestions and specific remarks to might help to improve the presentation of the study.

OUR REPLY: we thank the reviewer for the constructive comments on our manuscript.

General comments

(1) The presentation of the study is very taxon-centered. There are good examples of heritability and genetic correlations for behavioural and life-history traits in other mammals. Just doing it (again) for a primate species, does not in itself justifies a study. Given a large body of research on life-history and behaviour of this species (also in the field), it is a very suitable study system, but this needs to be clearer in the presentation.

OUR REPLY: Indeed, we based our study on a particular species that is of interest for primate research as well as for personality research. We added justifications based on the need (1) to take advantage of this unique colony of grey mouse lemur. This captive population has been founded several decades ago and disposes of data on life history traits that can be exploited in relation to personality. In the future, this study could represent a good reference for similar studies on wild mouse lemurs.

(2) The study is based on data from a captive colony. Captive populations are often built from a small founder population, which might cause some inbreeding effects, captivity effects, unintended directional selection for some traits etc. In the current presentation of their data the authors do not discuss any of these issues inherent of captive populations and I would urge them to do so in the discussion.

OUR REPLY: We added a paragraph in the discussion. We have good reason to assume that inbreeding is not higher in our population than in the wild, where breeding distance and homozygosity relatively high (Fredsted et al. 2005). Moreover, colony managers pay attention to avoid related individuals to mate which is made possible by the large number of individuals. With respect to founder effects, wild populations are also likely to “suffer” from funder effects because of the fragmentation of the Malagasy forest. (Olivieri et al 2008)

(3) This is my main concern. The authors collected a wealth of interesting data and present their findings clearly. However, their results could be discussed more directly in the light of current theory on the evolution and maintenance of (co)variation in behavioural and life-history traits. For example, the pace-of-life syndrome hypothesis makes very clear directed predictions about genetic correlations between traits. Also the life-history trade-off hypothesis lends itself to frame the study.

OUR REPLY: Indeed, the pace-of-life syndrome is a particularly suitable framework to work with in this case. This has been integrated more into the introduction of the manuscript.

(4) It remains unclear how personality was quantified. The behavioural variables were quantified repeatedly but were they repeatable? What personality score did you use in the models (average?, predicted individual intercepts?)? The definition of personality is not correct.

OUR REPLY: Behavioral variables were previously tested for repeatability. We added an explanation in the methods and the estimate for each personality trait. We did not have to average behavioral variables as we used linear mixed models with individual identity as a random factor to account for pseudoreplication. “We used the two personality traits that were described in Zabolocki-Thomas et al. (2018) that both presented medium repeatability in this dataset (42).” (L.135-136).

(4) The English would need some editing.

OUR REPLY: we have paid special attention to wording in the revised version.

Specific comments

Abstract

L8: See above, the aim is not well justified. Just doing a study in a primate does not justify it.

OUR REPLY: We detailed the justification as follows:

“Yet, how the sources variance in these traits are decomposed remains unclear and need additional data, and whether genetic correlations with life-history traits have an influence on personality variations and other phenotypic traits remains poorly understood.” (L.9-12)

L20: Replace “in” with “on”.

OUR REPLY: The change was done. (L.23)

Introduction

Overall, I think the introduction could develop a more theoretical framework to develop the study aims (see above). Predictions and hypotheses are clear but very descriptive and not developed from theory.

OUR REPLY: we have tried to render the introduction more conceptual in the revised version and have linked our predictions and hypotheses directly to the POLS theory. (First paragraph of the introduction)

L36: What is consistent variability? Personality is not defined as consistent variability but as consistent among-individual differences. Also, personality does not imply that behaviours are “fixed”. I would urge the authors to go back to classical papers on personality (e.g., Dingemanse et al. 2010 TREE).

OUR REPLY: **We understand why the formulation was inappropriate. “Personality is generally defined as statistically repeatable behaviours over context and time between individuals. (L.60)**

L41: These are incorrect references for the statement. Please cite meta-analyses on fitness consequences, not opinion papers.

OUR REPLY: **This part of the introduction was rephrased.**

L58: This is true but a rather weak justification for the study. I would suggest to motivate the study based on theory.

OUR REPLY: **We added that it was also rarely conducted in the same population for several types of traits. We also base our paper on the “pace-of-life” hypothesis to introduce the study on correlations between these traits.**

Methods

Please add a section on the colony (number of generations in captivity, inbreeding, breeding scheme (who with whom?), number of founders etc.).

OUR REPLY: **Information was added in the methods, (a) and (b) sections.**

Behavioural tests

Please specify whether these tests were performed in nocturnal conditions (activity period of the species) or diurnal conditions. Are the behavioural variables repeatable at the population level?

OUR REPLY: **We added information on repeatability and added that tests were realized in day-light conditions. (135-137)**

Statistics

How did you treat the agitation scores in the models, they are ordinary data?

OUR REPLY: **As described in the methods in the phenotypic trait section, agitation scores ranged from zero to eight and are discrete data since they are the sum of one, two or three points. They were not transformed in the univariate models and scaled in the bivariate models and they were treated as a gaussian distribution.**

How did you treat the repeated behaviours in the models (average expression, random intercepts...)?

OUR REPLY: **As behavioral data were repeated, we used mixed models and added individual identity as a random factor.**

L143: Please add the error term to the equation.

OUR REPLY: **The error term was added. (L.197)**

Bivariate models:

Please express clearly in the text, which pairs of variables were combined.

OUR REPLY: **We added a sentence in the methods: “We tested for genetic correlation between emergence latency and agitation, emergence latency and radius length, emergence latency and birth weight, emergence latency and growth rate, and genetic correlation between agitation and radius length, agitation and birth weight, agitation growth rate, (see results, Table 2).”**

Results

L168: Since you have two terms for environmental variation, be explicit here.

OUR REPLY: **The “common” environment was added. (L.224)**

Discussion

As stated above, the presentation of the results of the study would gain much from linking it closer with the current theoretical framework. Your results are extremely interesting, e.g. before the pace-of-life syndrome hypothesis. I would suggest to completely overwork this discussion going away from a list-like presentation of sources of variation in the various traits and replace it with a discussion of your results on heritability and particularly of genetic correlation between the different traits.

OUR REPLY: **we have modified the discussion as suggested to put more emphasis on trait correlation in the light of the POLS hypothesis.**

Moreover, as stated above, please add a discussion of the captivity problem.

OUR REPLY: **we now discuss effects of captivity as requested. (L.334-357)**

References

The reference list needs serious proof-reading (species names in italic, missing page numbers, missing journal names, etc.).

OUR REPLY: **We have carefully revised the reference list. We thank the reviewer for pointing this out and apologize for the remaining mistakes.**

Tables and Figures

Fig. 1: not needed.

OUR REPLY: **1 was deleted.**

Reviewer: 3

The authors have conducted behavioural assessments and physical trait measurements on a captive and pedigreed population of Grey Mouse Lemurs. It is unusual to see studies like this in primates and their sample sizes are reasonable for a primate study. This paper is of interest

to readers but is not quite up to date in methods and discussions and is not detailed enough to be replicated.

OUR REPLY: **we thank the reviewer for the constructive comments on our paper.**

This paper is reasonably thorough but is behind the times in the following aspects:

- the calculation of heritability with fixed effects is problematic. Please read (Wilson 2008). In short, fixed effects account for variance in the model. A better approach is that of (de Villemereuil et al. 2018) to account for fixed effects in calculating heritability. The variance accounted for by fixed effects in the model is calculated and then included in the heritability calculation.

OUR REPLY: **We followed the “animal model guide” published by Wilson et al. 2010 to conduct our estimations, which is quite a reference in quantitative genetics and animal model analyses.**

- there are papers showing differences between captive and wild population derived repeatibilities. (Bell et al. 2009) review might be an early one but there are also blue/great tit papers on this topic (Herborn et al. 2010) and somewhere some theory on differences in repeatability between captivity and the wild (I can't remember where though!). I would have liked to have seen a more nuanced introduction and discussion that included points such as this and some of the other points raised below.

OUR REPLY: **This is now discussed in the revised discussion.**

- the authors can calculate standard errors on their heritability estimates using the pin() function. This would be preferable to estimates with no uncertainty.

OUR REPLY: **we thank the reviewer for the suggestion.**

- the assumptions behind building their model are unusual. The authors start at the assumption of heritability and add other factors including the permanent environment effect to this, according to the methods. The repeatability includes the heritability (roughly speaking, but there are caveats to that), so heritability is a portion of the greater whole. Or, another way to think about it, the calculation is the heritability of repeatability. Therefore, they would be better to start with the assumption of a permanent environment effect and add the pedigree second.

OUR REPLY: **We are not sure we entirely understand what the referee is getting at. We apologize if our way of running the model was not straightforward or if the description thereof was not clear.**

A few other points came up as I read through the paper. In the discussion, evolvability in a captive population is a nice idea, but it is a bit of a stretch to imply a link to wild populations in the discussion. Especially since the opportunity for selection and heritability depend on population size and are population specific (sorry, I forget the name of the person who has done theory work on the opportunity for selection and population size). The authors discuss some of the interests and issues on lines 304 – 311 but it comes across as a little speculative and could have more grounding in the literature. Line 239-241 also doesn't mention that

heritability and evolvability are population specific, so there is great difficulty in drawing the comparison here. It would be good if the authors are more tentative about this statement.

OUR REPLY: We agree with the reviewer and we have nuanced these statements in the discussion. We also explicitly state that it would be important to study wild populations to validate the observations of our study.

The first paragraph of the introduction is very interesting, though it feels slightly disjointed from the rest of the text at the moment.

OUR REPLY: we have revised the introduction and have tried to integrate this better into the whole of the study.

Line 36: this is a naughty (read:bad) way to describe personality! Especially before a paper that uses variance partitioning to explore animal personality and heritability. Animal personality is the consistency of behaviour within an individual relative to other individuals within the same population. Or, animal personality is defined statistically as a greater proportion of behavioural variance within a population being attributable to within-individual similarity than between-individual differences. There are loads of possibilities, but the current formulation misses the subtlety that animal personality is 1) a statistical concept 2) distinct from individual differences in intra-individual variability (more similar concept than animal personality to the current phrasing the authors have used!) (Stamps et al. 2012) 3) is a relative population-based measure. There is no quantifiable ‘personality trait’ in an individual animal, it is only by comparison that we arrive at a concept of personality.

OUR REPLY: We agree and changed the phrasing of the definition of personality. (L60-68)

If the authors could do a power analysis for their pedigree, this would make their paper more robust. I think (Germain et al. 2018) has one for the Mandarte island sparrows, and there is a simple one in (Winney et al. 2018).

OUR REPLY: given that we only had two to three potential fathers for each offspring and that mothers are known our pedigree is extremely robust. We detailed the mating system and the pedigree construction in the methods. (L.114)

In general, there is a lack of detail in the methods such that I could not re-do the analysis or make a full assessment of whether they were appropriate. The lack of detail includes:
- were covariance models run with repeated measures or not? If run with repeated measures, the calculation of covariance is different and the authors should check out the appendix of (Dingemanse et al. 2012).

OUR REPLY: we followed the tutorial by Wilson et al (2010) when setting up our models.

-it seems some subjects died during the study but this is not mentioned in the initial sample sizes lines 80-85.

OUR REPLY: We added to the methods: “Complete data were not available for all individuals which explains why the sample size for each phenotypic trait varies and is

different from the total number of individuals present in the study (N=486). Indeed, we could not collect all the phenotypic data at the same time in particular to avoid additional stress during the behavioural tests. Some individuals could not be tested for all traits as some died or were unavailable (when involved in reproduction for example).” (L.168-173)

-authors state they transform but not that this is to improve model residuals. Also the authors don't state their distribution, so whilst they are probably running LMM because it's asreml this needs to be stated.

OUR REPLY: indeed, we added this in the revised version.

-do the authors include repeated measures when they calculate evolvability?

OUR REPLY: yes, but we no longer use the term “evolvability” as such in the manuscript.

- authors need to give repeatability for their ‘personality traits’ to show that they are repeatable traits.

OUR REPLY: Repeatability was already assessed with those personality tests in this dataset. We added details about it in the methods.

- please supply residual covariances for multivariate models so that the covariances can be viewed in context.

OUR REPLY: Residual covariance were added in table 2.

- line 299 mentions inbreeding coefficients, which another population e.g. the Mandarte song sparrow population (Jane Reid’s papers) takes into account during their calculations of heritability. Do the authors need to consider this here?

OUR REPLY: we did not take this into account. Colony management is design to avoid inbreeding.

Confusing points:

-line 126 pedigrees look like a random factor but they aren't. Re-word for clarity.

OUR REPLY: changed to: “with a pedigree incorporated to quantify the additive genetic variance” (L.177)

-introduction and methods: it is confusing to introduce the 'five personality traits' in the introduction and then never link them to the traits you actually measure. There is also a lot of great work on confirming that these traits match our human-imposed labels (Carter et al. 2013). It would be good if the authors read these to get an understanding of the controversy over the categorisation of behaviours.

OUR REPLY: We deleted the reference to the five personality traits in the introduction. We discussed the link with these five traits in the discussion

-in table 1, is N for the number of animals? It would be worth giving overall sample size below if it is.

OUR REPLY: **Yes, it is the sample size for the model. It is mentioned in the table. (L.241)**

- line 14: define your type of heritability. Narrow-sense is mentioned once in the discussion.

OUR REPLY: **“narrow” sense heritability was added. (L.17)**

- line 41: the sentence beginning ‘as such’ does not follow on from the previous sentence.

OUR REPLY: **This sentence was deleted for clarity.**

- line 50: the reference used here is not the first occurrence of this type of study, read work by Peter Biro e.g. (Biro & Post 2008).

OUR REPLY: **The reference was changed.**

- line 59-60: speculation too strongly worded. Replace with ‘one of the reasons for this could be...’

OUR REPLY: **This part of the introduction was deleted.**

Line 109: ‘grabbing’ has bad connotations, choose another suitable word like catching.

OUR REPLY: **The change was done. (L.153)**

-line 258: ‘general theory’ is what theory?

OUR REPLY: **This part was rephrased.**

Bell, A. M., Hankison, S. J., & Laskowski, K. L. (2009). The repeatability of behaviour: a meta-analysis. *Animal Behaviour*, 77(4), 771–783.

<https://doi.org/10.1016/j.anbehav.2008.12.022>

Biro, P. A., & Post, J. R. (2008). Rapid depletion of genotypes with fast growth and bold personality traits from harvested fish populations. *Proceedings of the National Academy of Sciences of the United States of America*, 105(8), 2919–2922.

<https://doi.org/10.1073/pnas.0708159105>

Carter, A. J., Feeney, W. E., Marshall, H. H., Cowlshaw, G., & Heinsohn, R. (2013). Animal personality: what are behavioural ecologists measuring? *Biological Reviews*, 88(2), 465–475.

<https://doi.org/10.1111/bry.12007>

de Villemereuil, P., Morrissey, M. B., Nakagawa, S., & Schielzeth, H. (2018). Fixed-effect variance and the estimation of repeatabilities and heritabilities: issues and solutions. *Journal of Evolutionary Biology*, 31(4), 621–632. <https://doi.org/10.1111/jeb.13232>

Dingemanse, N. J., Dochtermann, N. a., & Nakagawa, S. (2012). Defining behavioural syndromes and the role of ‘syndrome deviation’ in understanding their evolution. *Behavioral Ecology and Sociobiology*, 66(11), 1543–1548. <https://doi.org/10.1007/s00265-012-1416-2>

Germain, R. R., Wolak, M. E., Reid, J. M., & Germain, R. R. (2018). Individual repeatability and heritability of divorce in a wild population. *Biology Letters*, 14, 20180061.

Herborn, K. a., Macleod, R., Miles, W. T. S., Schofield, A. N. B., Alexander, L., & Arnold, K. E. (2010). Personality in captivity reflects personality in the wild. *Animal Behaviour*, 79(4), 835–843. <https://doi.org/10.1016/j.anbehav.2009.12.026>

Stamps, J. a., Briffa, M., & Biro, P. a. (2012). Unpredictable animals: individual differences in intraindividual variability (IIV). *Animal Behaviour*, 83(6), 1325–1334.

<https://doi.org/10.1016/j.anbehav.2012.02.017>

Wilson, A. J. (2008). Why h^2 does not always equal V_A/V_P ? *Journal of Evolutionary Biology*, 21(3), 647–650. <https://doi.org/10.1111/j.1420-9101.2008.01500.x>

Winney, I. S., Schroeder, J., Nakagawa, S., Hsu, Y.-H., Simons, M. J. P., Sánchez-Tójar, A., ... Burke, T. (2018). Heritability and social brood effects on personality in juvenile and adult life-history stages in a wild passerine. *Journal of Evolutionary Biology*, 31(1), 75–87.

<https://doi.org/10.1111/jeb.13197>